# LEARNING CAUSAL SEMANTIC REPRESENTATION FOR OUT-OF-DISTRIBUTION PREDICTION

## ABSTRACT

Conventional supervised learning methods, especially deep ones, are found to be sensitive to out-of-distribution (OOD) examples, largely because the learned representation mixes the semantic factor with the variation factor due to their domain-specific correlation, while only the semantic factor *causes* the output. To address the problem, we propose a Causal Semantic Generative model (CSG) based on causality to model the two factors separately, and learn it on a single training domain for prediction without (OOD generalization) or with unsupervised data (domain adaptation) in a test domain. We prove that CSG identifies the semantic factor on the training domain, and the invariance principle of causality subsequently guarantees the boundedness of OOD generalization error and the success of adaptation. We also design novel and delicate learning methods for both effective learning and easy prediction, following the first principle of variational Bayes and the graphical structure of CSG. Empirical study demonstrates the effect of our methods to improve test accuracy for OOD generalization and domain adaptation.

## 1 INTRODUCTION

Deep learning has initiated a new era of artificial intelligence where the potential of machine learning models is greatly unleashed. Despite the great success, these methods heavily rely on the independently-and-identically-distributed (IID) assumption. This does not always perfectly hold in practice, and the prediction of output (label, response, outcome) $y$ may be saliently affected in out-of-distribution (OOD) cases, even from an essentially irrelevant change to the input (covariate) $x$, like a position shift or rotation of the object in an image, or a change of background, illumination or style (Shen et al., 2018; He et al., 2019; Arjovsky et al., 2019). These phenomena pose serious concerns on the robustness and trustworthiness of machine learning methods and severely impede them from risk-sensitive scenarios.

Looking into the problem, although deep learning models allow extracting abstract representation for prediction with their powerful approximation capacity, the representation may be overconfident in the correlation between semantic factors $s$ (*e.g.*, shape of an object) and variation factors $v$ (*e.g.*, background, illumination, object position). The correlation may be domain-specific and spurious, and may change drastically in a new environment. So it has become a desire to learn representation that separates semantics $s$ from variations $v$ (Cai et al., 2019; Ilse et al., 2019). Formally, the importance of this goal is that $s$ represents the *cause* of $y$. Causal relations better reflect the fundamental mechanisms of nature, bringing the merit to machine learning that they tend to be universal and *invariant* across domains (Schölkopf et al., 2012; Peters et al., 2017; Schölkopf, 2019), thus providing the most transferable and confident information to unseen domains. Causality has also been shown to lead to proper domain adaptation (Schölkopf et al., 2012; Zhang et al., 2013), lower adaptation cost and lighter catastrophic forgetting (Peters et al., 2016; Bengio et al., 2019; Ke et al., 2019).

In this work, we propose a Causal Semantic Generative model (CSG) for proper and robust OOD prediction, including OOD generalization and domain adaptation. Both tasks have supervised data from a *single* training domain, but domain adaptation has unsupervised test-domain data during learning, while OOD generalization has no test-domain data, including cases where queries come sequentially or adaptation is unaffordable. **(1)** We build the model by cautiously following the principle of causality, where we explicitly separate the latent variables into a (group of) semantic factor $s$ and a (group of) variation factor $v$. We prove that under appropriate conditions CSG identifies the

semantic factor by fitting training data, even in presence of an $s$-$v$ correlation. **(2)** By leveraging the causal invariance, we prove that a well-learned CSG is guaranteed to have a bounded OOD generalization error. The bound shows how causal mechanisms affect the error. **(3)** We develop a domain adaptation method using CSG and causal invariance, which suggests to fix the causal generative mechanisms and adapt the prior to the new domain. We prove the identification of the new prior and the benefit of adaptation. **(4)** To learn and adapt the model from data, we design novel and delicate reformulations of the Evidence Lower BOund (ELBO) objective following the graphical structure of CSG, so that the inference models required therein can also serve for prediction, and modeling and optimizing inference models in both domains can be avoided. To our best knowledge, our work is the *first* to identify semantic factor and leverage latent causal invariance for OOD prediction with guarantees. Empirical improvement in OOD performance and adaptation is demonstrated by experiments on multiple tasks including shifted MNIST and ImageCLEF-DA task.

## 2 RELATED WORK

There have been works that aim to leverage the merit of causality for OOD prediction. For OOD generalization, some works ameliorate discriminative models towards a causal behavior. Bahadori et al. (2017) introduce a regularizer that reweights input dimensions based on their approximated causal effects to the output, and Shen et al. (2018) reweight training samples by amortizing causal effects among input dimensions. They are extended to nonlinear cases (Bahadori et al., 2017; He et al., 2019) via linear-separable representations. Heinze-Deml & Meinshausen (2019) enforce inference invariance by minimizing prediction variance within each label-identity group. These methods introduce no additional modeling effort, but may also be limited to capture invariant causal mechanisms (they are non-generative) and may only behave quantitatively causal in the training domain.

For domain adaptation/generalization, methods are developed under various causal assumptions (Schölkopf et al., 2012; Zhang et al., 2013) or using learned causal relations (Rojas-Carulla et al., 2018; Magliacane et al., 2018). Zhang et al. (2013); Gong et al. (2016; 2018) also consider certain ways of mechanism shift. The considered causality is among directly observed variables, which may not be suitable for general data like image pixels where causality rather lies between data and conceptual latent factors (Lopez-Paz et al., 2017; Besserve et al., 2018; Kilbertus et al., 2018). To consider latent factors, there are domain adaptation (Pan et al., 2010; Baktashmotlagh et al., 2013; Ganin et al., 2016; Long et al., 2015; 2018) and generalization methods (Muandet et al., 2013; Shankar et al., 2018) that learn a representation with domain-invariant marginal distribution, and have achieved remarkable results. Nevertheless, Johansson et al. (2019); Zhao et al. (2019) point out that this invariance is neither sufficient nor necessary to identify the true semantics and lower the adaptation error (Supplement D). Moreover, these methods and invariance risk minimization (Arjovsky et al., 2019) also assume the invariance in the inference direction (*i.e.*, data $\rightarrow$ representation), which may not be as general as causal invariance in the generative direction (Section 3.2).

There are also generative methods for domain adaptation/generalization that model latent factors. Cai et al. (2019); Ilse et al. (2019) introduce a semantic factor and a domain-feature factor. They assume the two factors are independent in both the generative and inference models, which may not meet reality closely. They also do not adapt the prior for domain shift thus resort to inference invariance. Zhang et al. (2020) consider a partially observed manipulation variable, while assume its independence from the output in both the joint and posterior, and the adaptation is inconsistent with causal invariance. Atzmon et al. (2020) consider similar latent factors, but use the same (uniform) prior in all domains. These methods also do not show guarantees to identify their latent factors. Teshima et al. (2020) leverage causal invariance and adapt the prior, while also assume latent independence and do not separate the semantic factor. They require some supervised test-domain data, and their deterministic and invertible mechanism also indicates inference invariance. In addition, most domain generalization methods require multiple training domains, with exceptions (*e.g.*, Qiao et al., 2020) that still seek to augment domains. In contrast, CSG leverages causal invariance, and has guarantee to identify the semantic factor from a single training domain, even with a correlation to the variation factor.

Generative supervised learning is not new (Mcauliffe & Blei, 2008; Kingma et al., 2014), but most works do not consider the encoded causality. Other works consider solving causality tasks, notably causal/treatment effect estimation (Louizos et al., 2017; Yao et al., 2018; Wang & Blei, 2019). The task does not focus on OOD prediction, and requires labels for both treated and controlled groups.

Disentangling latent representations is also of interest in unsupervised learning. Despite some empirical success (Chen et al., 2016; Higgins et al., 2017; Chen et al., 2018), Locatello et al. (2019) conclude that it is impossible to guarantee the disentanglement in unsupervised settings. Khemakhem et al. (2019; 2020) show an encouraging result that disentangled representation can be identified up to a permutation with a cause of the latent variable observed. But the methods cannot separate the semantic factor from variation for supervised learning, and require observing sufficiently many different values of the cause variable, making it hard to leverage labels.

Causality with latent variable has been considered in a rich literature (Verma & Pearl, 1991; Spirtes et al., 2000; Richardson et al., 2002; Hoyer et al., 2008; Shpitser et al., 2014), while most works focus on the consequence on observation-level causality. Others consider identifying the latent variable. Janzing et al. (2009); Lee et al. (2019) show the identifiability under additive noise or similar assumptions. For discrete data, a "simple" latent variable can be identified under various specifications (Janzing et al., 2011; Sgouritsa et al., 2013; Kocaoglu et al., 2018). Romeijn & Williamson (2018) leverage interventional datasets. Over these works, we step further to separate and identify the latent variable as semantic and variation factors, and show the benefit for OOD prediction.

## 3   THE CAUSAL SEMANTIC GENERATIVE MODEL

To develop the model seriously and soberly based on causality, we require the formal definition of causality: *two variables have a causal relation, denoted as "cause→effect", if externally intervening the cause (by changing variables out of the considered system) may change the effect, but not vice versa* (Pearl, 2009; Peters et al., 2017). We then follow the logic below to build our model. [1]

**(1)** It may be a general case that neither $y \rightarrow x$ (*e.g.*, adding noise to the labels in a dataset does not change the images) nor $x \rightarrow y$ holds (*e.g.*, intervening an image by *e.g.* breaking a camera sensor unit when taking the image, does not change how the photographer labels it), as also argued by Peters et al. (2017, Section 1.4); Kilbertus et al. (2018). So we employ a generative model (*i.e.*, not only modeling $p(y|x)$), and introduce a latent variable $z$ to capture factors with causal relations.

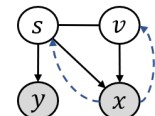

Figure 1: The graphical structure of the proposed Causal Semantic Generative model (CSG) for the semantic $s$ and variation $v$ latent factors and supervised data $(x, y)$. Black solid arrows represent invariant causal generating mechanisms $p(x|s, v)$ and $p(y|s)$, the black undirected edge represent a domain-specific prior $p(s, v)$, and blue dashed bended arrows represent the inference model $q(s, v|x)$ for learning and prediction.

**(2)** The latent variable $z$ as underlying generating factors (*e.g.*, object features like shape and texture, background and illumination in imaging) is plausible to cause both $x$ (*e.g.*, the change of object shape or background makes a different image, but breaking a camera sensor unit does not change the object shape or background) and $y$ (*e.g.*, the photographer would give a different label if the object shape, texture, *etc.* had been replaced by those of a different object, but noise-corrupting the label does not change the object features). So we orient the edges in the generative direction $z \rightarrow (x, y)$, as also adopted by Mcauliffe & Blei (2008); Peters et al. (2017); Teshima et al. (2020). This is in contrast to Cai et al. (2019); Ilse et al. (2019; 2020); Castro et al. (2020) who treat $y$ as the cause of a semantic factor, which, when $y$ is also a noisy observation, makes unreasonable implications (*e.g.*, adding noise to the labels in a dataset automatically changes object features and consequently the images, and changing the object features does not change the label). This difference is also discussed by Peters et al. (2017, Section 1.4); Kilbertus et al. (2018).

**(3)** We attribute all $x$-$y$ relation to the existence of some latent factors ("purely common cause", Lee et al., 2019; Janzing et al., 2009), and exclude $x$-$y$ edges. This can be achieved as long as $z$ holds sufficient information of data (*e.g.*, with shape, background *etc.* fixed, breaking a sensor unit does not change the label, and noise-corrupting the label does not change the image). Promoting this restriction reduces arbitrariness in explaining $x$-$y$ relation and benefits the identification of $z$. This is in contrast to Kingma et al. (2014); Zhang et al. (2020); Castro et al. (2020) who treat $y$ as a cause of $x$ since no latent variable is introduced between.

---

[1]Supplement C provides more explanations on the model.

**(4)** Not all latent factors are the causes of $y$ (*e.g.*, changing the shape may alter the label, while changing the background does not). We thus split the latent variable as $z = (s, v)$ and remove the edge $v \to y$, where $s$ represents the *semantic* factor of $x$ that causes $y$, and $v$ describes the *variation* or diversity in generating $x$. This formalizes the intuition on the concepts in Introduction.

**(5)** The variation $v$ often has a relation to the semantics $s$, which is often a spurious correlation (*e.g.*, desks prefer a workspace background, but they can also appear in bedrooms and beds can also appear in workspace). So we keep the undirected $s$-$v$ edge. Although $v$ is not a cause of $y$, modeling it explicitly is worth the effort since otherwise it would still be implicitly incorporated in $s$ anyway through the $s$-$v$ correlation. We summarize these conclusions in the following definition.

**Definition 3.1** (CSG). A *Causal Semantic Generative Model* (CSG) $p = (p(s, v), p(x|s, v), p(y|s))$ is a generative model on data variables $x \in \mathcal{X} \subset \mathbb{R}^{d_{\mathcal{X}}}$ and $y \in \mathcal{Y}$ with semantic $s \in \mathcal{S} \subset \mathbb{R}^{d_{\mathcal{S}}}$ and variation $v \in \mathcal{V} \subset \mathbb{R}^{d_{\mathcal{V}}}$ latent variables, following the graphical structure shown in Fig. 1.

### 3.1 The Causal Invariance Principle

The domain-invariance of causal relations translates to the following principle for CSG:

**Principle 3.2** (causal invariance). The causal generative mechanisms $p(x|s, v)$ and $p(y|s)$ in CSG are invariant across domains, and the change of prior $p(s, v)$ is the only source of domain shift.

It is supported by the invariance of basic laws of nature (Schölkopf et al., 2012; Peters et al., 2017; Besserve et al., 2018; Bühlmann, 2018; Schölkopf, 2019). Other works instead introduce domain index (Cai et al., 2019; Ilse et al., 2019; 2020; Castro et al., 2020) or manipulation variables (Zhang et al., 2020; Khemakhem et al., 2019; 2020) to model distribution change explicitly. They require multiple training domains or additional observations, and such changes can also be explained under causal invariance as long as the latent variable includes all shifted factors (*e.g.*, domain change of images can be attributed to a different preference of shape, style, texture, background, *etc.* and their correlations, while the processes generating image and label from them remain the same).

### 3.2 Comparison with Inference Invariance

Domain-invariant-representation-based adaptation and generalization methods, and invariant risk minimization (Arjovsky et al., 2019) for domain generalization, use a shared feature extractor across domains. This effectively assumes the invariance of the process in the other direction, *i.e.*, inferring the latent representation from data. We note that in its supportive examples (*e.g.*, inferring the object position from an image, or extracting the fundamental frequency from a vocal audio), generating mechanisms are nearly deterministic and invertible, so that the posterior is almost determined by the inverse function, and causal invariance implies inference invariance. For noisy or degenerate mechanisms (Fig. 2), ambiguity occurs during inference since there may be multiple values of a latent feature that generate the same observation. The inferred feature would notably rely on the prior through the Bayes rule. Since the prior changes across domains, the inference rule then changes by nature, which challenges the existence of a domain-shared feature extractor. In this case, causal invariance is more reliable than inference invariance.

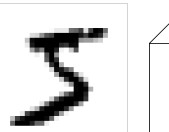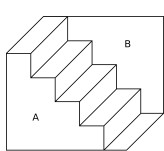

Figure 2: Examples of noisy (left) or degenerate (right) generating mechanisms that lead to ambiguity in inference. Left: handwritten digit that may be generated as either "3" or "5". Right: Schröder's stairs that may be generated with either A or B being the nearer surface. Inference results notably rely on the prior on the digits/surfaces, which is domain-specific.

To leverage causal invariance, we adjust the prior conservatively for OOD generalization (CSG-ind) and data-driven for domain adaptation (CSG-DA), so together with the invariant generative mechanisms, it gives a different and more reliable inference rule than that following inference invariance.

## 4 Method

We develop learning, adaptation and prediction methods for OOD generalization and domain adaptation using CSG following the causal invariance Principle 3.2, and devise practical objectives using variational Bayes. Supplement E.1 details all the derivations.

### 4.1 METHOD FOR OOD GENERALIZATION

For OOD generalization, a CSG $p = (p(s, v), p(x|s, v), p(y|s))$ needs to first learn from the supervised data from an underlying data distribution $p^*(x, y)$ on the training domain. Maximizing likelihood $\mathbb{E}_{p^*(x,y)}[\log p(x, y)]$ is intractable since $p(x, y)$ given by the CSG $p$ is hard to estimate effectively. We thus adopt the Evidence Lower BOund (ELBO) $\mathcal{L}_{q,p}(x, y) := \mathbb{E}_{q(s,v|x,y)}[\log \frac{p(s,v,x,y)}{q(s,v|x,y)}]$ (Jordan et al., 1999; Wainwright et al., 2008) as a tractable surrogate, which requires an auxiliary inference model $q(s, v|x, y)$ to estimate the expectation effectively. Maximizing $\mathcal{L}_{q,p}$ w.r.t $q$ drives $q$ towards the posterior $p(s, v|x, y)$ and meanwhile makes $\mathcal{L}_{q,p}$ a tighter lower bound of $\log p(x, y)$. The expected ELBO $\mathbb{E}_{p^*(x,y)}[\mathcal{L}_{q,p}(x, y)]$ then drives $p(x, y)$ towards $p^*(x, y)$.

However, the subtlety with supervised learning is that after fitting data, evaluating $p(y|x)$ for prediction is still hard. We thus propose to employ a model for $q(s, v, y|x)$ instead. The required inference model can be then expressed as $q(s, v|x, y) = q(s, v, y|x)/q(y|x)$ where $q(y|x) = \int q(s, v, y|x) \, \mathrm{d}s\mathrm{d}v$. It reformulates the expected ELBO as:

$$\mathbb{E}_{p^*(x,y)}[\mathcal{L}_{q,p}(x, y)] = \mathbb{E}_{p^*(x)}\mathbb{E}_{p^*(y|x)}[\log q(y|x)] + \mathbb{E}_{p^*(x)}\mathbb{E}_{q(s,v,y|x)}\Big[\frac{p^*(y|x)}{q(y|x)}\log\frac{p(s, v, x, y)}{q(s, v, y|x)}\Big]. \quad (1)$$

The first term is the common cross entropy loss (negative) driving $q(y|x)$ towards $p^*(y|x)$. Once this is achieved, the second term becomes the expected ELBO $\mathbb{E}_{p^*(x)}[\mathcal{L}_{q(s,v,y|x),p}(x)]$ that drives $q(s, v, y|x)$ towards $p(s, v, y|x)$ (and $p(x)$ towards $p^*(x)$). Since the target $p(s, v, y|x)$ admits the factorization $p(s, v|x)p(y|s)$ (since $(v, x) \perp\!\!\!\perp y|s$ ) where $p(y|s)$ is already given by the CSG, we can further ease the modeling of $q(s, v, y|x)$ as $q(s, v|x)p(y|s)$. The ELBO is then reformulated as:

$$\mathcal{L}_{q,p}(x, y) = \log q(y|x) + \frac{1}{q(y|x)}\mathbb{E}_{q(s,v|x)}\Big[p(y|s)\log\frac{p(s, v)p(x|s, v)}{q(s, v|x)}\Big], \quad (2)$$

where $q(y|x) = \mathbb{E}_{q(s,v|x)}[p(y|s)]$. The CSG $p$ and $q(s, v|x)$ are to be optimized. The expectations can be estimated by Monte Carlo, and their gradients can be estimated using the reparameterization trick (Kingma & Welling, 2014). When well optimized, $q(s, v|x)$ well approximates $p(s, v|x)$, so $q(y|x)$ then well approximates $p(y|x) = \mathbb{E}_{p(s,v|x)}[p(y|s)]$ for prediction.

**CSG-ind**    To actively mitigate the spurious $s$-$v$ correlation from the training domain, we also consider a CSG with an **ind**ependent prior $p^{\perp\!\!\!\perp}(s, v) := p(s)p(v)$ for prediction in the unknown test domain, where $p(s)$ and $p(v)$ are the marginals of $p(s, v)$. The independent prior $p^{\perp\!\!\!\perp}(s, v)$ encourages the model to stay neutral on the $s$-$v$ correlation. It has a larger entropy than $p(s, v)$ (Cover & Thomas, 2006, Theorem 2.6.6), so it reduces the information of the training-domain-specific prior. The model then relies more on the invariant generative mechanisms, thus better leverages causal invariance and gives more reliable prediction than that following inference invariance.

For the method, note that the prediction is given by $p^{\perp\!\!\!\perp}(y|x) = \mathbb{E}_{p^{\perp\!\!\!\perp}(s,v|x)}[p(y|s)]$, so we use an inference model for $q^{\perp\!\!\!\perp}(s, v|x)$ that approximates $p^{\perp\!\!\!\perp}(s, v|x)$. However, learning on the training domain still requires the original inference model $q(s, v|x)$. To save the cost of building and learning two inference models, we propose to use $q^{\perp\!\!\!\perp}(s, v|x)$ to represent $q(s, v|x)$. Noting that their targets are related by $p(s, v|x) = \frac{p(s,v)}{p^{\perp\!\!\!\perp}(s,v)}\frac{p^{\perp\!\!\!\perp}(x)}{p(x)}p^{\perp\!\!\!\perp}(s, v|x)$, we formulate $q(s, v|x) = \frac{p(s,v)}{p^{\perp\!\!\!\perp}(s,v)}\frac{p^{\perp\!\!\!\perp}(x)}{p(x)}q^{\perp\!\!\!\perp}(s, v|x)$ accordingly, so that this $q(s, v|x)$ achieves its target once $q^{\perp\!\!\!\perp}(s, v|x)$ does. The ELBO then becomes:

$$\mathcal{L}_{q,p}(x, y) = \log \pi(y|x) + \frac{1}{\pi(y|x)}\mathbb{E}_{q^{\perp\!\!\!\perp}(s,v|x)}\Big[\frac{p(s, v)}{p^{\perp\!\!\!\perp}(s, v)}p(y|s)\log\frac{p^{\perp\!\!\!\perp}(s, v)p(x|s, v)}{q^{\perp\!\!\!\perp}(s, v|x)}\Big], \quad (3)$$

where $\pi(y|x) := \mathbb{E}_{q^{\perp\!\!\!\perp}(s,v|x)}\Big[\frac{p(s,v)}{p^{\perp\!\!\!\perp}(s,v)}p(y|s)\Big]$. The CSG $p$ and $q(s, v|x)$ are to be optimized (note that $p^{\perp\!\!\!\perp}(s, v)$ is determined by $p(s, v)$ in the CSG $p$). Prediction is given by $p^{\perp\!\!\!\perp}(y|x) \approx \mathbb{E}_{q^{\perp\!\!\!\perp}(s,v|x)}[p(y|s)]$.

### 4.2 METHOD FOR DOMAIN ADAPTATION

When unsupervised data is available from an underlying data distribution $\tilde{p}^*(x)$ on the test domain, we can leverage it for adaptation. According to the causal invariance Principle 3.2, we only need to adapt for the test-domain prior $\tilde{p}(s, v)$ and the corresponding inference model $\tilde{q}(s, v|x)$, while the causal mechanisms $p(x|s, v)$ and $p(y|s)$ are not optimized. Adaptation is done by fitting the test

data via maximizing $\mathbb{E}_{\tilde{p}^*(x)}[\mathcal{L}_{\tilde{q},\tilde{p}}(x)]$, where the ELBO is in the standard form:

$$\mathcal{L}_{\tilde{q},\tilde{p}}(x) = \mathbb{E}_{\tilde{q}(s,v|x)}\Big[\log\Big(\tilde{p}(s,v)p(x|s,v)/\tilde{q}(s,v|x)\Big)\Big]. \tag{4}$$

Prediction is given by $\tilde{p}(y|x) \approx \mathbb{E}_{\tilde{q}(s,v|x)}[p(y|s)]$. Similar to the case of CSG-ind, we need $\tilde{q}(s,v|x)$ for prediction, but $q(s,v|x)$ is still required for learning on the training domain. When data from both domains are available during learning, we can save the effort of modeling and learning $q(s,v|x)$ using a similar technique. We formulate it using $\tilde{q}(s,v|x)$ as $q(s,v|x) = \frac{\tilde{p}(x)}{p(x)}\frac{p(s,v)}{\tilde{p}(s,v)}\tilde{q}(s,v|x)$ following the same relation between their targets, and the ELBO on the training domain becomes:

$$\mathcal{L}_{q,p}(x,y) = \log \pi(y|x) + \frac{1}{\pi(y|x)}\mathbb{E}_{\tilde{q}(s,v|x)}\Big[\frac{p(s,v)}{\tilde{p}(s,v)}p(y|s)\log\frac{\tilde{p}(s,v)p(x|s,v)}{\tilde{q}(s,v|x)}\Big], \tag{5}$$

where $\pi(y|x) := \mathbb{E}_{\tilde{q}(s,v|x)}\big[\frac{p(s,v)}{\tilde{p}(s,v)}p(y|s)\big]$. The CSG $p$ and $\tilde{q}(s,v|x)$ are to be optimized (not for $\tilde{p}(s,v)$). The resulting method, termed CSG-DA, solves both optimizations (4, 5) simultaneously.

For implementing the three methods, note that only one inference model is required in each case. Supplement E.2 shows its implementation from a general discriminative model (*e.g.*, how to select its hidden nodes as $s$ and $v$). In practice $x$ often has a much larger dimension than $y$, making the supervised part of the training-domain ELBO (*i.e.*, the first term in its formulation Eq. (1)) scales smaller than the unsupervised part. So we include an additional cross entropy loss in the objectives.

## 5 THEORY

We now establish guarantee for the methods on identifying the semantic factor and the subsequent merits for OOD generalization and domain adaptation. We only consider the infinite-data regime to isolate another source of error from finite data. Supplement A shows all the proofs. Identifiability is hard to achieve for latent variable models (Koopmans & Reiersol, 1950; Murphy, 2012; Yacoby et al., 2019; Locatello et al., 2019), since it is a task beyond modeling observational relations (Janzing et al., 2009; Peters et al., 2017). Assumptions are required to draw definite conclusions.

**Assumption 5.1** (additive noise). There exist nonlinear functions $f$ and $g$ with bounded derivatives up to third-order, and independent random variables $\mu$ and $\nu$, such that $p(x|s,v) = p_\mu(x - f(s,v))$, and $p(y|s) = p_\nu(y - g(s))$ for continuous $y$ or $p(y|s) = \text{Cat}(y|g(s))$ for categorical $y$.

This structure disables describing a bivariate joint distribution in both generating directions (Zhang & Hyvärinen (2009, Theorem 8), Peters et al. (2014, Proposition 23)), and is widely adopted in directed causal discovery (Janzing et al., 2009; Bühlmann et al., 2014). CSG needs this since it should make the causal direction exclusive. It is also easy to implement with deep models (Kingma & Welling, 2014), so does not essentially restrict model capacity.

**Assumption 5.2** (bijectivity). Function $f$ is bijective and $g$ is injective.

It is a common assumption for identifiability (Janzing et al., 2009; Shalit et al., 2017; Khemakhem et al., 2019; Lee et al., 2019). Under Assumption 5.1, it is a sufficient condition (Peters et al., 2014, Proposition 17; Peters et al., 2017, Proposition 7.4) of causal minimality (Peters et al., 2014, p.2012; Peters et al., 2017, Definition 6.33), a fundamental requirement for identifiability (Peters et al., 2014, Proposition 7; Peters et al., 2017, p.109). Particularly, $s$ and $v$ are otherwise allowed to have dummy dimensions that $f$ and $g$ simply ignore, raising another ambiguity against identifiability. On the other hand, according to the commonly acknowledged manifold hypothesis (Weinberger & Saul, 2006; Fefferman et al., 2016) that data tends to lie on a lower-dimensional manifold embedded in the data space, we can take $\mathcal{X}$ as the manifold and such a bijection exists as a coordinate map, which is an injection to the original data space (thus allowing $d_\mathcal{S} + d_\mathcal{V} < d_\mathcal{X}$).

### 5.1 IDENTIFIABILITY THEORY

We first formalize the goal of identifying the semantic factor.

**Definition 5.3** (semantic-equivalence). We say two CSGs $p$ and $p'$ are *semantic-equivalent*, if there exists a homeomorphism[2] $\Phi$ on $\mathcal{S} \times \mathcal{V}$, such that **(i)** its output dimensions in $\mathcal{S}$ is constant of $v$:

---

[2]A transformation is a homeomorphism if it is a continuous bijection with continuous inverse.

$\Phi^{\mathcal{S}}(s, v) = \Phi^{\mathcal{S}}(s)$ for any $v \in \mathcal{V}$, and **(ii)** it acts as a *reparameterization* from $p$ to $p'$: $\Phi_{\#}[p_{s,v}] = p'_{s,v}$, $p(x|s, v) = p'(x|\Phi(s, v))$ and $p(y|s) = p'(y|\Phi^{\mathcal{S}}(s))$.

It is an equivalent relation if $\mathcal{V}$ is connected and is either open or closed in $\mathbb{R}^{d_{\mathcal{V}}}$ (Supplement A.1). Here, $\Phi_{\#}[p_{s,v}]$ denotes the pushed-forward distribution[3] by $\Phi$, *i.e.* the distribution of the transformed random variable $\Phi(s, v)$ when $(s, v) \sim p_{s,v}$. As a reparameterization, $\Phi$ allows the two models to have different latent-variable parameterizations while inducing the same distribution on the observed data variables $(x, y)$ (Supplement Lemma A.2). At the heart of the definition, the $v$-constancy of $\Phi^{\mathcal{S}}$ implies that $\Phi$ is *semantic-preserving*: one model *does not mix* the other's $v$ into its $s$, so that the $s$ variables of both models hold equivalent information.

We say that a learned CSG $p$ *identifies* the semantic factor if it is semantic-equivalent to the ground-truth CSG $p^*$. This identification cannot be characterized by the *statistical independence* between $s$ and $v$ (as in Cai et al. (2019); Ilse et al. (2019); Zhang et al. (2020)), which is not sufficient (Locatello et al., 2019) nor necessary (due to the existence of spurious correlation). Another related concept is *disentanglement*. It requires that a semantic transformation on $x$ changes the learned $s$ only (Higgins et al., 2018; Besserve et al., 2020), while the identification here does not require the learned $v$ to be constant of the ground-truth $s$.

To identify the semantic factor, the ground-truth model could at most provide its information via the data distribution $p^*(x, y)$. Although semantic-equivalent CSGs induce the same distribution on $(x, y)$, the inverse is nontrivial. The following theorem shows that the semantic-identifiability can be achieved under appropriate conditions.

**Theorem 5.4** (semantic-identifiability). *With Assumptions 5.1 and 5.2, a well-learned CSG $p$ with $p(x, y) = p^*(x, y)$ is semantic-equivalent to the ground-truth CSG $p^*$, if $\log p(s, v)$ and $\log p^*(s, v)$ have bounded derivatives up to the second-order, and that[4]* **(i)** $\frac{1}{\sigma_{\mu}^2} \to \infty$ *where $\sigma_{\mu}^2 := \mathbb{E}[\mu^{\top}\mu]$, or* **(ii)** $p_{\mu}$ *has an a.e. non-zero characteristic function (e.g., a Gaussian distribution).*

**Remarks.** **(1)** The requirement on $p(s, v)$ and $p^*(s, v)$ excludes *extreme* training data that show a deterministic $s$-$v$ relation, which makes the $(s, v)$ density functions unbounded and discontinuous. In that case (*e.g.*, all desks appear in workspace and all beds in bedrooms), one cannot tell whether the label $y$ is caused by $s$ (*e.g.*, the shape) or by $v$ (*e.g.*, the background).

**(2)** In condition **(i)**, $\frac{1}{\sigma_{\mu}^2}$ measures the *intensity* of the causal mechanism $p(x|s, v)$. A strong $p(x|s, v)$ helps disambiguating values of $(s, v)$ in generating a given $x$. The condition makes $p(x|s, v)$ so strong that it is almost deterministic and invertible, so inference invariance also holds (Section 3.2). Supplement A.2 provides a quantitative reference of large intensity for a practical consideration, and Supplement B gives a non-asymptotic extension showing how the intensity trades-off the tolerance of equalities in Definition 5.3. Condition **(ii)** covers more than inference invariance. It roughly implies that different values of $(s, v)$ a.s. produce different distributions $p(x|s, v)$ on $\mathcal{X}$, so their roles in generating $x$ become clear which helps identification.

**(3)** The theorem does not contradict the impossibility result by Locatello et al. (2019), which considers disentangling each latent dimension with an unconstrained $(s, v) \to (x, y)$, while we identify $s$ as a whole with the edge $v \to y$ removed which breaks the $s$-$v$ symmetry.

### 5.2 OOD GENERALIZATION THEORY

The causal invariance Principle 3.2 forms the ground-truth CSG on the test domain as $\tilde{p}^* = (\tilde{p}^*(s, v), p^*(x|s, v), p^*(y|s))$ with the new ground-truth prior $\tilde{p}^*(s, v)$, which gives the optimal predictor $\tilde{\mathbb{E}}^*[y|x]$ [5] on the test domain. The principle also leads to the invariance of identified causal mechanisms, which shows that the OOD generalization error of a CSG is bounded:

**Theorem 5.5** (OOD generalization error). *With Assumptions 5.1 and 5.2, for a semantically-identified CSG $p$ on the training domain with reparameterization $\Phi$, we have up to $O(\sigma_{\mu}^2)$ that*

---

[3]The definition of $\Phi_{\#}[p_{s,v}]$ requires $\Phi$ to be measurable. This is satisfied by the continuity of $\Phi$ as a homeomorphism (as long as the considered $\sigma$-field is the Borel $\sigma$-field) (Billingsley, 2012, Theorem 13.2).

[4]To be precise, the semantic-equivalent conclusions are that the equalities in Definition 5.3 hold asymptotically in the limit $\frac{1}{\sigma_{\mu}^2} \to \infty$ for condition **(i)**, and hold a.e. for condition **(ii)**.

[5]For categorical $y$, the expectation of $y$ is taken under the one-hot representation.

*for any* $x \in \mathrm{supp}(p_x) \cap \mathrm{supp}(\tilde{p}_x^*)$,

$$\left| \mathbb{E}[y|x] - \tilde{\mathbb{E}}^*[y|x] \right| \leqslant \sigma_\mu^2 \|\nabla g(s)\|_2 \left\| J_{f^{-1}}(x) \right\|_2^2 \|\nabla \log(p(s,v)/\tilde{p}(s,v))\|_2 \Big|_{(s,v)=f^{-1}(x)}, \quad (6)$$

*where* $\mathrm{supp}$ *denotes the support of a distribution,* $J_{f^{-1}}$ *is the Jacobian matrix of* $f^{-1}$, *and* $\tilde{p}_{s,v} := \Phi_\#[\tilde{p}_{s,v}^*]$ *is the test-domain prior under the parameterization of the identified CSG* $p$. [6]

The result shows that when the causal mechanism $p(x|s, v)$ is strong, especially in the extreme case $\sigma_\mu = 0$ where inference invariance also holds, it dominates prediction over the prior and the generalization error diminishes. In more general cases where only causal invariance holds, the prior change deviates the prediction rule. The prior-change term $\|\nabla \log(p(s,v)/\tilde{p}(s,v))\|_2$ measures the *hardness* or severity of OOD. It diminishes in IID cases, and makes the bound lose its effect when the two priors do not share their support. Using a CSG to fit training data enforces causal invariance and other assumptions, so its $\mathbb{E}[y|x]$ behaves more faithfully in low $p^*(x)$ area and the boundedness becomes more plausible in practice. CSG-ind further actively uses an independent prior whose larger support covers more $\tilde{p}_{s,v}$ candidates.

### 5.3 DOMAIN ADAPTATION THEORY

In cases of weak causal mechanism or violent prior change, the new ground-truth prior $p_{s,v}^*$ is important for prediction. The domain adaptation method learns a new prior $\tilde{p}_{s,v}$ by fitting unsupervised test-domain data, with causal mechanisms shared. Once the mechanisms are identified, $p_{s,v}^*$ can also be identified under the learned parameterization, and prediction can be made precise.

**Theorem 5.6** (domain adaptation error). *Under the conditions of Theorem 5.4, for a semantically-identified CSG* $p$ *on the training domain with reparameterization* $\Phi$, *if its new prior* $\tilde{p}_{s,v}$ *for the test domain is well-learned with* $\tilde{p}(x) = \tilde{p}^*(x)$, *then* $\tilde{p}_{s,v} = \Phi_\#[\tilde{p}_{s,v}^*]$, *and* $\tilde{\mathbb{E}}[y|x] = \tilde{\mathbb{E}}^*[y|x]$ *for any* $x \in \mathrm{supp}(\tilde{p}_x^*)$.

Different from existing domain adaptation bounds (Supplement D), Theorems 5.5 and 5.6 allow different inference models in the two domains, thus go beyond inference invariance.

## 6 EXPERIMENTS

For baselines of OOD generalization, apart from the conventional supervised learning optimizing cross entropy (CE), we also consider a causal discriminative method CNBB (He et al., 2019), and a generative method supervised VAE (sVAE) which is a counterpart of CSG that does not separate its latent variable into $s$ and $v$. For domain adaptation, we consider well-acknowledged DANN (Ganin et al., 2016), DAN (Long et al., 2015) and CDAN (Long et al., 2018) methods implemented in the `dalib` package (Jiang et al., 2020), and also sVAE using a similar method as CSG-DA. All methods share the same optimization setup. We align the scale of the CE term in the objectives of all methods, and tune their hyperparameters to lie on the margin that makes the final accuracy near 1 on a validation set from the training domain. See Supplement F for details.

### 6.1 SHIFTED MNIST

We consider an OOD prediction task on MNIST to classify digits "0" and "1". In the training data, "0"s are horizontally shifted at random by $\delta$ pixels with $\delta \sim \mathcal{N}(-5, 1^2)$, and "1"s by $\delta \sim \mathcal{N}(5, 1^2)$ pixels. We consider two test domains where the digits are not moved, or are shifted $\delta \sim \mathcal{N}(0, 2^2)$ pixels. Both domains have balanced classes. We implement all methods using multilayer perceptron which is not naturally shift invariant. We use a larger architecture for discriminative and domain adaptation methods to compensate the additional generative components of generative methods.

The OOD performance is shown in Table 1. For OOD generalization, CSG gives more genuine predictions in unseen domains, thanks to the identification of the semantic factor. CSG-ind performs even better, demonstrating the merit of approaching a CSG with an independent prior. Other methods are more significantly misled by the position factor from the spurious correlation. CNBB ameliorates the position bias, but not as thoroughly without explicit structures for causal mechanisms. CSG

---

[6]The 2-norm $\|\cdot\|_2$ for matrices refers to the induced operator norm (not the Frobenius norm).

Table 1: Accuracy (%) of various methods (ours in bold) on OOD generalization (left) and domain adaptation (right) for shifted MNIST. Averaged over 10 runs.

| shift | CE | CNBB | sVAE | **CSG** | **CSG-ind** | DANN | DAN | CDAN | sVAE-DA | **CSG-DA** |
|---|---|---|---|---|---|---|---|---|---|---|
| $= 0$ | 53.3± 8.8 | 74.4± 6.2 | 60.5±13.9 | 90.9± 6.8 | **94.5± 4.5** | 91.6± 5.5 | 11.3±11.5 | 60.1±41.4 | 50.3± 3.9 | **95.2±11.9** |
| $\mathcal{N}(0,2^2)$ | 52.7± 2.8 | 58.0± 1.7 | 58.6± 5.8 | 64.8± 2.7 | **66.5± 4.7** | 47.0± 2.9 | 50.1± 2.9 | 49.2± 5.6 | 68.0± 7.5 | **76.0± 3.4** |

Table 2: Results on ImageCLEF-DA (ima, 2014). Results of CE, DANN, DAN and CDAN are taken from Long et al. (2018).

| task | CE | CNBB | sVAE | **CSG** | **CSG-ind** | DANN | DAN | CDAN | sVAE-DA | **CSG-DA** |
|---|---|---|---|---|---|---|---|---|---|---|
| **C→P** | 65.5± 0.3 | 72.7± 1.1 | 73.3± 1.0 | 73.4± 0.8 | **73.7± 0.6** | 74.3± 0.5 | 69.2± 0.4 | **74.5± 0.3** | 74.3± 0.4 | 74.4± 0.6 |
| **P→C** | 91.2± 0.3 | 91.7± 0.2 | 91.6± 0.9 | 92.3± 0.5 | **92.5± 0.3** | 91.5± 0.6 | 89.8± 0.4 | **93.5± 0.4** | 92.6± 0.3 | 92.9± 0.3 |

also outperforms sVAE, showing the benefit of separating semantics and variation and modeling the variation explicitly, so the model could consciously drive semantic representation into $s$. For domain adaptation, existing methods differ a lot, and are hard to perform well on both test domains. When fail to identify, adaptation sometimes even worsens the result, as the misleading representation based on position gets strengthened on the unsupervised test data. CSG is benefited from adaptation by leveraging test data in a proper way that identifies the semantics.

## 6.2 IMAGECLEF-DA

ImageCLEF-DA (ima, 2014) is a standard benchmark dataset for the ImageCLEF 2014 domain adaptation challenge. We select a pair of adaptation tasks between two of its domains: **C**altech-256 and **P**ascal VOC 2012. Each domain has 12 classes and 600 images following a different distribution from each other. We adopt the same setup as in Long et al. (2018), including the ResNet50 structure (He et al., 2016) pretrained on ImageNet as the backbone of the discriminative/inference model. For generative methods, we leverage the DCGAN generator (Radford et al., 2015) pretrained on Cifar10.

Table 2 shows the results. We see that CSG(-ind) achieves the best OOD generalization result, and performs comparable with modern domain adaptation methods. On this task, the underlying causal mechanism may be very noisy (*e.g.*, photos taken from inside and outside both count for the aircraft class), making identification hard. So CSG-DA does not make a salient improvement.

## 7 CONCLUSION AND DISCUSSION

We tackle OOD generalization and domain adaptation tasks by proposing a Causal Semantic Generative model (CSG), which builds upon a causal reasoning, and models semantic and variation factors separately while allowing their correlation. Using the invariance principle of causality, we develop effective and delicate methods for learning, adaptation and prediction, and prove the identification of the semantic factor, the boundedness of OOD generalization error, and the success of adaptation under appropriate conditions. Experiments show the improved performance in both tasks.

The consideration of separating semantics from variation extends to broader examples regarding robustness. Convolutional neural networks are found to change its prediction under a different texture but the same shape (Geirhos et al., 2019; Brendel & Bethge, 2019). Adversarial vulnerability (Szegedy et al., 2014; Goodfellow et al., 2015; Kurakin et al., 2016) extends variation factors to human-imperceptible features, *i.e.* the adversarial noise, which is shown to have a strong spurious correlation with semantics (Ilyas et al., 2019). The separation also matters for fairness when a sensitive variation factor may change prediction due to a spurious correlation. Our methods are potentially beneficial in these examples.

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

## SUPPLEMENTARY MATERIALS

## A PROOFS

We first introduce some handy concepts and results to make the proof succinct. We begin with extended discussions on CSG.

**Definition A.1.** A homeomorphism $\Phi$ on $\mathcal{S} \times \mathcal{V}$ is called a *reparameterization* from CSG $p$ to CSG $p'$, if $\Phi_{\#}[p_{s,v}] = p'_{s,v}$, and $p(x|s,v) = p'(x|\Phi(s,v))$ and $p(y|s) = p'(y|\Phi^{\mathcal{S}}(s,v))$ for any $(s,v) \in \mathcal{S} \times \mathcal{V}$. A reparameterization $\Phi$ is called to be *semantic-preserving*, if its output dimensions in $\mathcal{S}$ is constant of $v$: $\Phi^{\mathcal{S}}(s,v) = \Phi^{\mathcal{S}}(s)$ for any $v \in \mathcal{V}$.

Note that a reparameterization unnecessarily has its output dimensions in $\mathcal{S}$, *i.e.* $\Phi^{\mathcal{S}}(s,v)$, constant of $v$. The condition that $p(y|s) = p'(y|\Phi^{\mathcal{S}}(s,v))$ for any $v \in \mathcal{V}$ does not indicate that $\Phi^{\mathcal{S}}(s,v)$ is constant of $v$, since $p'(y|s')$ may ignore the change of $s' = \Phi^{\mathcal{S}}(s,v)$ from the change of $v$. The following lemma shows the meaning of a reparameterization: it allows a CSG to vary while inducing the same distribution on the observed data variables $(x,y)$ (*i.e.*, holding the same effect on describing data).

**Lemma A.2.** *If there exists a reparameterization $\Phi$ from CSG $p$ to CSG $p'$, then $p(x,y) = p'(x,y)$.*

*Proof.* By the definition of a reparameterization, we have:

$$p(x,y) = \int p(s,v)p(x|s,v)p(y|s)\,\mathrm{d}s\mathrm{d}v = \int \Phi_{\#}^{-1}[p'_{s,v}](s,v)p'(x|\Phi(s,v))p'(y|\Phi^{\mathcal{S}}(s,v))\,\mathrm{d}s\mathrm{d}v$$

$$= \int p'_{s,v}(s',v')p'(x|s',v')p'(y|s')\,\mathrm{d}s'\mathrm{d}v' = p'(x,y),$$

where we used variable substitution $(s',v') := \Phi(s,v)$ in the second-last equality. Note that by the definition of pushed-forward distribution and the bijectivity of $\Phi$, $\Phi_{\#}[p_{s,v}] = p'_{s,v}$ implies $p_{s,v} = \Phi_{\#}^{-1}[p'_{s,v}]$, and $\int f(s',v')p'_{s,v}(s',v')\,\mathrm{d}s'\mathrm{d}v' = \int f(\Phi(s,v))\Phi_{\#}^{-1}[p'_{s,v}](s,v)\,\mathrm{d}s\mathrm{d}v$ (can also be verified deductively using the rule of change of variables, *i.e.* Lemma A.4 in the following). $\qquad\square$

The definition of semantic-equivalence (Definition 5.3) can be rephrased by the *existence* of a semantic-preserving reparameterization. With appropriate model assumptions, we can show that *any* reparameterization between two CSGs is semantic-preserving, so that semantic-preserving CSGs cannot be converted to each other by a reparameterization that mixes $s$ with $v$.

**Lemma A.3.** *For two CSGs $p$ and $p'$, if $p'(y|s)$ has a statistics $M'(s)$ that is an injective function of $s$, then* any *reparameterization $\Phi$ from $p$ to $p'$, if exists, has its $\Phi^{\mathcal{S}}$ constant of $v$.*

*Proof.* Let $\Phi = (\Phi^{\mathcal{S}}, \Phi^{\mathcal{V}})$ be any reparameterization from $p$ to $p'$. Then the condition that $p(y|s) = p'(y|\Phi^{\mathcal{S}}(s,v))$ for any $v \in \mathcal{V}$ indicates that $M(s) = M'(\Phi^{\mathcal{S}}(s,v))$. If there exist $s \in \mathcal{S}$ and $v^{(1)} \neq v^{(2)} \in \mathcal{V}$ such that $\Phi^{\mathcal{S}}(s,v^{(1)}) \neq \Phi^{\mathcal{S}}(s,v^{(2)})$, then $M'(\Phi^{\mathcal{S}}(s,v^{(1)})) \neq M'(\Phi^{\mathcal{S}}(s,v^{(2)}))$ since $M'$ is injective. This violates $M(s) = M'(\Phi^{\mathcal{S}}(s,v))$ which requires both $M'(\Phi^{\mathcal{S}}(s,v^{(1)}))$ and $M'(\Phi^{\mathcal{S}}(s,v^{(2)}))$ to be equal to $M(s)$. So $\Phi^{\mathcal{S}}(s,v)$ must be constant of $v$. $\qquad\square$

We then introduce two mathematical facts.

**Lemma A.4** (rule of change of variables)**.** *Let $z$ be a random variable on a Euclidean space $\mathbb{R}^{d_z}$ with density function $p_z(z)$, and let $\Phi$ be a homeomorphism on $\mathbb{R}^{d_z}$ whose inverse $\Phi^{-1}$ is differentiable. Then the distribution of the transformed random variable $z' = \Phi(z)$ has a density function $\Phi_{\#}[p_z](z') = p_z(\Phi^{-1}(z'))|J_{\Phi^{-1}}(z')|$, where $|J_{\Phi^{-1}}(z')|$ denotes the absolute value of the determinant of the Jacobian matrix $(J_{\Phi^{-1}}(z'))_{ia} := \frac{\partial}{\partial z'_i}(\Phi^{-1})_a(z')$ of $\Phi^{-1}$ at $z'$.*

*Proof.* See *e.g.*, Billingsley (2012, Theorem 17.2). Note that a homeomorphism is (Borel) measurable since it is continuous (Billingsley, 2012, Theorem 13.2), so the definition of $\Phi_{\#}[p_z]$ is valid. $\qquad\square$

**Lemma A.5.** *Let $\mu$ be a random variable whose characteristic function is a.e. non-zero. For two functions $f$ and $f'$ on the same space, we have: $f * p_\mu = f' * p_\mu \iff f = f'$ a.e., where $(f * p_\mu)(x) := \int f(x)p_\mu(x - \mu)\,\mathrm{d}\mu$ denotes convolution.*

*Proof.* The function equality $f * p_\mu = f' * p_\mu$ leads to the equality under Fourier transformation $\mathscr{F}[f * p_\mu] = \mathscr{F}[f' * p_\mu]$, which gives $\mathscr{F}[f]\mathscr{F}[p_\mu] = \mathscr{F}[f']\mathscr{F}[p_\mu]$. Since $\mathscr{F}[p_\mu]$ is the characteristic function of $p_\mu$, the condition that it is a.e. non-zero indicates that $\mathscr{F}[f] = \mathscr{F}[f']$ a.e. thus $f = f'$ a.e. See also Khemakhem et al. (2019, Theorem 1). $\qquad\square$

### A.1 PROOF OF THE EQUIVALENCE RELATION

**Proposition A.6.** *The semantic-equivalence defined in Definition 5.3 is an equivalence relation if $\mathcal{V}$ is connected and is either open or closed in $\mathbb{R}^{d_\mathcal{V}}$.*

*Proof.* Let $\Phi$ be a semantic-preserving reparameterization from one CSG $p = (p(s, v), p(x|s, v), p(y|s))$ to another $p' = (p'(s, v), p'(x|s, v), p'(y|s))$. It has its $\Phi^{\mathcal{S}}$ constant of $v$, so we can write $\Phi(s, v) = (\Phi^{\mathcal{S}}(s), \Phi^{\mathcal{V}}(s, v)) =: (\phi(s), \psi_s(v))$.

**(1)** We first show that $\phi$, and $\psi_s$ for any $s \in \mathcal{S}$, are homeomorphisms on $\mathcal{S}$ and $\mathcal{V}$, respectively, and that $\Phi^{-1}(s', v') = (\phi^{-1}(s'), \psi_{\phi^{-1}(s')}^{-1}(v'))$.

- Since $\Phi(\mathcal{S} \times \mathcal{V}) = \mathcal{S} \times \mathcal{V}$, so $\phi(\mathcal{S}) = \Phi^{\mathcal{S}}(\mathcal{S}) = \mathcal{S}$, so $\phi$ is surjective.
- Suppose that there exists $s' \in \mathcal{S}$ such that $\phi^{-1}(s') = \{s^{(i)}\}_{i \in \mathcal{I}}$ contains multiple distinct elements.
  1. Since $\Phi$ is surjective, for any $v' \in \mathcal{V}$, there exist $i \in \mathcal{I}$ and $v \in \mathcal{V}$ such that $(s', v') = \Phi(s^{(i)}, v) = (\phi(s^{(i)}), \psi_{s^{(i)}}(v))$, which means that $\bigcup_{i \in \mathcal{I}} \psi_{s^{(i)}}(\mathcal{V}) = \mathcal{V}$.
  2. Since $\Phi$ is injective, the sets $\{\psi_{s^{(i)}}(\mathcal{V})\}_{i \in \mathcal{I}}$ must be mutually disjoint. Otherwise, there would exist $i \neq j \in \mathcal{I}$ and $v^{(1)}, v^{(2)} \in \mathcal{V}$ such that $\psi_{s^{(i)}}(v^{(1)}) = \psi_{s^{(j)}}(v^{(2)})$ thus $\Phi(s^{(i)}, v^{(1)}) = (s', \psi_{s^{(i)}}(v^{(1)})) = (s', \psi_{s^{(j)}}(v^{(2)})) = \Phi(s^{(j)}, v^{(2)})$, which violates the injectivity of $\Phi$ since $s^{(i)} \neq s^{(j)}$.
  3. In the case where $\mathcal{V}$ is open, then so is any $\psi_{s^{(i)}}(\mathcal{V}) = \Phi(s^{(i)}, \mathcal{V})$ since $\Phi$ is continuous. But the union of disjoint open sets $\bigcup_{i \in \mathcal{I}} \psi_{s^{(i)}}(\mathcal{V}) = \mathcal{V}$ cannot be connected. This violates the condition that $\mathcal{V}$ is connected.
  4. A similar argument holds in the case where $\mathcal{V}$ is closed.
  So $\phi^{-1}(s')$ contains only one unique element for any $s' \in \mathcal{S}$. So $\phi$ is injective.
- The above argument also shows that for any $s' \in \mathcal{S}$, we have $\bigcup_{i \in \mathcal{I}} \psi_{s^{(i)}}(\mathcal{V}) = \psi_{\phi^{-1}(s')}(\mathcal{V}) = \mathcal{V}$. For any $s \in \mathcal{S}$, there exists $s' \in \mathcal{S}$ such that $s = \phi^{-1}(s')$, so we have $\psi_s(\mathcal{V}) = \mathcal{V}$. So $\psi_s$ is surjective for any $s \in \mathcal{S}$.
- Suppose that there exist $v^{(1)} \neq v^{(2)} \in \mathcal{V}$ such that $\psi_s(v^{(1)}) = \psi_s(v^{(2)})$. Then $\Phi(s, v^{(1)}) = (\phi(s), \psi_s(v^{(1)})) = (\phi(s), \psi_s(v^{(2)})) = \Phi(s, v^{(2)})$, which contradicts the injectivity of $\Phi$ since $v^{(1)} \neq v^{(2)}$. So $\psi_s$ is injective for any $s \in \mathcal{S}$.
- That $\Phi$ is continuous and $\Phi(s, v) = (\phi(s), \psi_s(v))$ indicates that $\phi$ and $\psi_s$ are continuous. For any $(s', v') \in \mathcal{S} \times \mathcal{V}$, we have $\Phi(\phi^{-1}(s'), \psi_{\phi^{-1}(s')}^{-1}(v')) = (\phi(\phi^{-1}(s')), \psi_{\phi^{-1}(s')}(\psi_{\phi^{-1}(s')}^{-1}(v'))) = (s', v')$. Applying $\Phi^{-1}$ to both sides gives $\Phi^{-1}(s', v') = (\phi^{-1}(s'), \psi_{\phi^{-1}(s')}^{-1}(v'))$.
- Since $\Phi^{-1}$ is continuous, $\phi^{-1}$ and $\psi_s^{-1}$ are also continuous.

**(2)** We now show that the relation is an equivalence relation. It amounts to showing the following three properties.

- Reflexivity. For two identical CSGs, we have $p(s, v) = p'(s, v)$, $p(x|s, v) = p'(x|s, v)$ and $p(y|s) = p'(y|s)$. So the identity map as $\Phi$ obviously satisfies all the requirements.
- Symmetry. Let $\Phi$ be a semantic-preserving reparameterization from $p = (p(s, v), p(x|s, v), p(y|s))$ to $p' = (p'(s, v), p'(x|s, v), p'(y|s))$. From the above conclusion in **(1)**, we know that $(\Phi^{-1})^{\mathcal{S}}(s', v') = \phi^{-1}(s')$ is semantic-preserving. Also, $\Phi^{-1}$ is a homeomorphism on $\mathcal{S} \times \mathcal{V}$ since $\Phi$ is. So we only need to show that $\Phi^{-1}$ is a reparameterization from $p'$ to $p$ for symmetry.

1. From the definition of pushed-forward distribution, we have $\Phi_\#^{-1}[p'_{s,v}] = p_{s,v}$ if $\Phi_\#[p_{s,v}] = p'_{s,v}$. It can also be verified through the rule of change of variables (Lemma A.4) when $\Phi$ and $\Phi^{-1}$ are differentiable. From $\Phi_\#[p_{s,v}] = p'_{s,v}$, we have for any $(s', v')$, $p_{s,v}(\Phi^{-1}(s', v'))|J_{\Phi^{-1}}(s', v')| = p'_{s,v}(s', v')$. Since for any $(s, v)$ there exists $(s', v')$ such that $(s, v) = \Phi^{-1}(s', v')$, this implies that for any $(s, v)$, $p_{s,v}(s, v)|J_{\Phi^{-1}}(\Phi(s, v))| = p'_{s,v}(\Phi(s, v))$, or $p_{s,v}(s, v) = p'_{s,v}(\Phi(s, v))/|J_{\Phi^{-1}}(\Phi(s, v))| = p'_{s,v}(\Phi(s, v))|J_\Phi(s, v)|$ (inverse function theorem), which means that $p_{s,v} = \Phi_\#^{-1}[p'_{s,v}]$ by the rule of change of variables.

2. For any $(s', v')$, there exists $(s, v)$ such that $(s', v') = \Phi(s, v)$, so $p'(x|s', v') = p'(x|\Phi(s, v)) = p(x|s, v) = p(x|\Phi^{-1}(s', v'))$, and $p'(y|s') = p'(y|\Phi^{\mathcal{S}}(s)) = p(y|s) = p(y|(\Phi^{-1})^{\mathcal{S}}(s'))$.

So $\Phi^{-1}$ is a reparameterization from $p'$ to $p$.

- Transitivity. Given a third CSG $p'' = (p''(s, v), p''(x|s, v), p''(y|s))$ that is semantic-equivalent to $p'$, there exists a semantic-preserving reparameterization $\Phi'$ from $p'$ to $p''$. It is easy to see that $(\Phi' \circ \Phi)^{\mathcal{S}}(s, v) = \Phi'^{\mathcal{S}}(\Phi^{\mathcal{S}}(s, v)) = \Phi'^{\mathcal{S}}(\Phi^{\mathcal{S}}(s))$ is constant of $v$ thus semantic-preserving. As the composition of two homeomorphisms $\Phi$ and $\Phi'$ on $\mathcal{S} \times \mathcal{V}$, $\Phi' \circ \Phi$ is also a homeomorphism. So we only need to show that $\Phi' \circ \Phi$ is a reparameterization from $p$ to $p''$ for transitivity.

  1. From the definition of pushed-forward distribution, we have $(\Phi' \circ \Phi)_\#[p_{s,v}] = \Phi'_\#[\Phi_\#[p_{s,v}]] = \Phi'_\#[p'_{s,v}] = p''_{s,v}$ if $\Phi_\#[p_{s,v}] = p'_{s,v}$ and $\Phi'_\#[p'_{s,v}] = p''_{s,v}$. It can also be verified through the rule of change of variables (Lemma A.4) when $\Phi^{-1}$ and $\Phi'^{-1}$ are differentiable. For any $(s'', v'')$, we have

$$(\Phi' \circ \Phi)_\#[p_{s,v}](s'', v'') = p_{s,v}((\Phi' \circ \Phi)^{-1}(s'', v''))|J_{(\Phi' \circ \Phi)^{-1}}(s'', v'')|$$

$$= p_{s,v}(\Phi^{-1}(\Phi'^{-1}(s'', v'')))|J_{\Phi^{-1}}(\Phi'^{-1}(s'', v''))||J_{\Phi'^{-1}}(s'', v'')|$$

$$= \Phi_\#[p_{s,v}](\Phi'^{-1}(s'', v''))|J_{\Phi'^{-1}}(s'', v'')|$$

$$= p'_{s,v}(\Phi'^{-1}(s'', v''))|J_{\Phi'^{-1}}(s'', v'')| = \Phi'_\#[p'_{s,v}](s'', v'') = p''_{s,v}(s'', v'').$$

  2. For any $(s, v)$, we have:

$$p(x|s, v) = p'(x|\Phi(s, v)) = p''(x|\Phi'(\Phi(s, v))) = p''(x|(\Phi' \circ \Phi)(s, v)),$$

$$p(y|s) = p'(y|\Phi^{\mathcal{S}}(s)) = p''(y|\Phi'^{\mathcal{S}}(\Phi^{\mathcal{S}}(s))) = p''(y|(\Phi' \circ \Phi)^{\mathcal{S}}(s)).$$

So $\Phi' \circ \Phi$ is a reparameterization from $p$ to $p''$.

This completes the proof for an equivalence relation. $\qquad\square$

## A.2 Proof of the Semantic-Identifiability Theorem 5.4

We present a more general and detailed version of Theorem 5.4 and prove it. The theorem in the main context corresponds to conclusions **(ii)** and **(i)** below by taking the two CSGs $p'$ and $p$ as the well-learned $p$ and the ground-truth CSGs $p^*$, respectively.

**Theorem 5.4'** (semantic-identifiability). *Consider CSGs $p$ and $p'$ that have Assumptions 5.1 and 5.2 hold, with the bounded derivative conditions specified to be that for both CSGs, $f^{-1}$ and $g$ are twice and $f$ thrice differentiable with mentioned derivatives bounded. Further assume that their priors have bounded densities and their $\log p(s, v)$ have bounded derivatives up to the second-order. If the two CSGs have $p(x, y) = p'(x, y)$, then they are semantic-equivalent, under the conditions that:* [7]
*(i) $p_\mu$ has an a.e. non-zero characteristic function (e.g., a Gaussian distribution);*
*(ii) $\frac{1}{\sigma_\mu^2} \to \infty$, where $\sigma_\mu^2 := \mathbb{E}[\mu^\top \mu]$;*
*(iii) $\frac{1}{\sigma_\mu^2} \gg B'^2_{f^{-1}} \max\{B'_{\log p} B'_g + \frac{1}{2} B''_g + \frac{3}{2} d B'_{f^{-1}} B''_f B'_g, B_p B'^d_{f^{-1}}(B'^2_{\log p} + B''_{\log p} + 3 d B'_{f^{-1}} B''_f B'_{\log p} + 3 d^{\frac{3}{2}} B'^2_{f^{-1}} B''^2_f + d^3 B'''_f B'_{f^{-1}})\}$, where $d := d_{\mathcal{S}} + d_{\mathcal{V}}$, and for both CSGs, the constant $B_p$ bounds $p(s, v)$, $B'_{f^{-1}}, B'_g, B'_{\log p}$ and $B''_f, B''_g, B''_{\log p}$ bound the 2-norms[8] of the gradient/Jacobian and the Hessians of the respective functions, and $B'''_f$ bounds all the 3rd-order derivatives of $f$.*

---

[7] To be precise, the conclusions are that the equalities in Definition 5.3 hold a.e. for condition **(i)**, hold asymptotically in the limit $\frac{1}{\sigma_\mu^2} \to \infty$ for condition **(ii)**, and hold up to a negligible quantity for condition **(iii)**.

[8] As an induced operator norm for matrices (not the Frobenius norm).

*Proof.* Without loss of generality, we assume that $\mu$ and $\nu$ (for continuous $y$) have zero mean. If it is not, we can redefine $f(s,v) := f(s,v) + \mathbb{E}[\mu]$ and $\mu := \mu - \mathbb{E}[\mu]$ (similarly for $\nu$ for continuous $y$) which does not alter the joint distribution $p(s,v,x,y)$ nor violates any assumptions. Also without loss of generality, we consider one scalar component (dimension) $l$ of $y$, and abuse the use of symbols $y$ and $g$ for $y_l$ and $g_l$ to avoid unnecessary complication. Note that for continuous $y$, due to the additive noise structure $y = g(s) + \nu$ and that $\nu$ has zero mean, we also have $\mathbb{E}[y|s] = g(s)$ as the same as the categorical $y$ case (under the one-hot representation). We sometimes denote $z := (s, v)$ for convenience.

First note that for both CSGs and both continuous and categorical $y$, by construction $g(s)$ is a sufficient statistics of $p(y|s)$ (not only the expectation $\mathbb{E}[y|s]$), and it is injective. So by Lemma A.3, we only need to show that there exists a reparameterization from $p$ to $p'$. We will show that $\Phi := f'^{-1} \circ f$ is such a reparameterization.

Since $f$ and $f'$ are bijective and continuous, we have $\Phi^{-1} = f^{-1} \circ f'$, so $\Phi$ is bijective and $\Phi$ and $\Phi^{-1}$ are continuous. So $\Phi$ is a homeomorphism. Also, by construction, we have:

$$p(x|z) = p_\mu(x - f(z)) = p_\mu(x - f'(f'^{-1}(f(z)))) = p_\mu(x - f'(\Phi(z))) = p'(x|\Phi(z)). \quad (7)$$

So we only need to show that $p(x,y) = p'(x,y)$ indicates $\Phi_\#[p_z] = p'_z$ and $p(y|s) = p'(y|\Phi^\mathcal{S}(s,v)), \forall v \in \mathcal{V}$ under the conditions.

**Proof under condition (i).** We begin with a useful reformulation of the integral $\int t(z)p(x|z)\,\mathrm{d}z$ for a general function $t$ of $z$. We will encounter integrals in this form. By Assumption 5.1, we have $p(x|z) = p_\mu(x - f(z))$, so we consider a transformation $\Psi_x(z) := x - f(z)$ and let $\mu = \Psi_x(z)$. It is invertible, $\Psi_x^{-1}(\mu) = f^{-1}(x - \mu)$, and $J_{\Psi_x^{-1}}(\mu) = -J_{f^{-1}}(x - \mu)$. By these definitions and the rule of change of variables, we have:

$$\int t(z)p(x|z)\,\mathrm{d}z = \int t(z)p_\mu(\Psi_x(z))\,\mathrm{d}z = \int t(\Psi_x^{-1}(\mu))p(\mu)\left|J_{\Psi_x^{-1}}(\mu)\right|\mathrm{d}\mu$$

$$= \int t(f^{-1}(x-\mu))p(\mu)\left|J_{f^{-1}}(x-\mu)\right|\mathrm{d}\mu$$

$$= \mathbb{E}_{p(\mu)}[(\bar{t}V)(x-\mu)] \quad (8)$$

$$= (f_\#[t] * p_\mu)(x), \quad (9)$$

where we have denoted functions $\bar{t} := t \circ f^{-1}$, $V := \left|J_{f^{-1}}\right|$, and abused the push-forward notation $f_\#[t]$ for a general function $t$ to formally denote $(t \circ f^{-1})\left|J_{f^{-1}}\right| = \bar{t}V$.

According to the graphical structure of CSG, we have:

$$p(x) = \int p(z)p(x|z)\,\mathrm{d}z, \quad (10)$$

$$\mathbb{E}[y|x] = \frac{1}{p(x)}\int yp(x,y)\,\mathrm{d}y = \frac{1}{p(x)}\iint yp(z)p(x|z)p(y|s)\,\mathrm{d}z\mathrm{d}y$$

$$= \frac{1}{p(x)}\int p(z)p(x|z)\mathbb{E}[y|s]\,\mathrm{d}z = \frac{1}{p(x)}\int g(s)p(z)p(x|z)\,\mathrm{d}z. \quad (11)$$

So from Eq. (9), we have:

$$p(x) = (f_\#[p_z] * p_\mu)(x), \qquad \mathbb{E}[y|x] = \frac{1}{p(x)}(f_\#[gp_z] * p_\mu)(x). \quad (12)$$

Matching the data distribution $p(x,y) = p'(x,y)$ indicates both $p(x) = p'(x)$ and $\mathbb{E}[y|x] = \mathbb{E}'[y|x]$. Using Lemma A.5 under condition (i), this further indicates $f_\#[p_z] = f'_\#[p'_z]$ a.e. and $f_\#[gp_z] = f'_\#[g'p'_z]$ a.e. The former gives $\Phi_\#[p_z] = p'_z$. The latter can be reformed as $\bar{g}f_\#[p_z] = \bar{g}'f'_\#[p'_z]$ a.e., so $\bar{g} = \bar{g}'$ a.e., where we have denoted $\bar{g} := g \circ (f^{-1})^\mathcal{S}$ and $\bar{g}' := g' \circ (f'^{-1})^\mathcal{S}$ similarly. From $\bar{g} = \bar{g}'$, we have for any $v \in \mathcal{V}$,

$$g(s) = g((f^{-1} \circ f)^\mathcal{S}(s,v)) = g((f^{-1})^\mathcal{S}(f(s,v))) = \bar{g}(f(s,v))$$

$$= \bar{g}'(f(s,v)) = g'((f'^{-1})^\mathcal{S}(f(s,v))) = g'(\Phi^\mathcal{S}(s,v)). \quad (13)$$

For both continuous and categorical $y$, $g(s)$ uniquely determines $p(y|s)$. So the above equality means that $p(y|s) = p'(y|\Phi^\mathcal{S}(s,v))$ for any $v \in \mathcal{V}$.

**Proof under condition (ii).** Applying Eq. (8) to Eqs. (10, 11), we have:

$$p(x) = \mathbb{E}_{p(\mu)}[(\bar{p}_z V)(x - \mu)], \qquad \mathbb{E}[y|x] = \frac{1}{p(x)}\mathbb{E}_{p(\mu)}[(\bar{g}\bar{p}_z V)(x - \mu)],$$

where we have similarly denoted $\bar{p}_z := p_z \circ f^{-1}$. Under condition **(ii)**, $\mathbb{E}[\mu^\top \mu]$ is infinitesimal, so we can expand the expressions w.r.t $\mu$. For $p(x)$, we have:

$$p(x) = \mathbb{E}_{p(\mu)}\left[\bar{p}_z V - \nabla(\bar{p}_z V)^\top \mu + \frac{1}{2}\mu^\top \nabla\nabla^\top(\bar{p}_z V)\mu + O(\mathbb{E}[\|\mu\|_2^3])\right]$$

$$= \bar{p}_z V + \frac{1}{2}\mathbb{E}_{p(\mu)}\left[\mu^\top \nabla\nabla^\top(\bar{p}_z V)\mu\right] + O(\sigma_\mu^3),$$

where all functions are evaluated at $x$. For $\mathbb{E}[y|x]$, we first expand $1/p(x)$ using $\frac{1}{x+\varepsilon} = \frac{1}{x} - \frac{\varepsilon}{x^2} + O(\varepsilon^2)$ to get: $\frac{1}{p(x)} = \frac{1}{\bar{p}_z V} - \frac{1}{2\bar{p}_z^2 V^2}\mathbb{E}_{p(\mu)}\left[\mu^\top \nabla\nabla^\top(\bar{p}_z V)\mu\right] + O(\sigma_\mu^3)$. The second term is expanded as: $\bar{g}\bar{p}_z V + \frac{1}{2}\mathbb{E}_{p(\mu)}\left[\mu^\top \nabla\nabla^\top(\bar{g}\bar{p}_z V)\mu\right] + O(\sigma_\mu^3)$. Combining the two parts, we have:

$$\mathbb{E}[y|x] = \bar{g} + \frac{1}{2}\mathbb{E}_{p(\mu)}\left[\mu^\top\left((\nabla\log\bar{p}_z V)\nabla\bar{g}^\top + \nabla\bar{g}(\nabla\log\bar{p}_z V)^\top + \nabla\nabla^\top\bar{g}\right)\mu\right] + O(\sigma_\mu^3). \quad (14)$$

This equation holds for any $x \in \text{supp}(p_x)$ since the expectation is taken w.r.t the distribution $p(x, y)$; in other words, the considered $x$ here is any value generated by the model. So up to $O(\sigma_\mu^2)$,

$$|p(x) - (\bar{p}_z V)(x)| = \frac{1}{2}\left|\mathbb{E}_{p(\mu)}\left[\mu^\top \nabla\nabla^\top(\bar{p}_z V)\mu\right]\right| \leqslant \frac{1}{2}\mathbb{E}_{p(\mu)}\left[\left|\mu^\top \nabla\nabla^\top(\bar{p}_z V)\mu\right|\right]$$

$$\leqslant \frac{1}{2}\mathbb{E}_{p(\mu)}\left[\|\mu\|_2\left\|\nabla\nabla^\top(\bar{p}_z V)\right\|_2\|\mu\|_2\right] = \frac{1}{2}\mathbb{E}[\mu^\top\mu]\left\|\nabla\nabla^\top(\bar{p}_z V)\right\|_2$$

$$= \frac{1}{2}\mathbb{E}[\mu^\top\mu]|\bar{p}_z V|\left\|\nabla\nabla^\top\log\bar{p}_z V + (\nabla\log\bar{p}_z V)(\nabla\log\bar{p}_z V)^\top\right\|_2$$

$$\leqslant \frac{1}{2}\mathbb{E}[\mu^\top\mu]|\bar{p}_z V|\left(\left\|\nabla\nabla^\top\log\bar{p}_z V\right\|_2 + \left\|\nabla\log\bar{p}_z V\right\|_2^2\right), \quad (15)$$

$$|\mathbb{E}[y|x] - \bar{g}(x)| = \frac{1}{2}\left|\mathbb{E}_{p(\mu)}\left[\mu^\top\left((\nabla\log\bar{p}_z V)\nabla\bar{g}^\top + \nabla\bar{g}(\nabla\log\bar{p}_z V)^\top + \nabla\nabla^\top\bar{g}\right)\mu\right]\right|$$

$$\leqslant \frac{1}{2}\mathbb{E}_{p(\mu)}\left[\left|\mu^\top\left((\nabla\log\bar{p}_z V)\nabla\bar{g}^\top + \nabla\bar{g}(\nabla\log\bar{p}_z V)^\top + \nabla\nabla^\top\bar{g}\right)\mu\right|\right]$$

$$\leqslant \frac{1}{2}\mathbb{E}_{p(\mu)}\left[\|\mu\|_2\left\|(\nabla\log\bar{p}_z V)\nabla\bar{g}^\top + \nabla\bar{g}(\nabla\log\bar{p}_z V)^\top + \nabla\nabla^\top\bar{g}\right\|_2\|\mu\|_2\right]$$

$$\leqslant \frac{1}{2}\mathbb{E}[\mu^\top\mu]\left(\left\|(\nabla\log\bar{p}_z V)\nabla\bar{g}^\top\right\|_2 + \left\|\nabla\bar{g}(\nabla\log\bar{p}_z V)^\top\right\|_2 + \left\|\nabla\nabla^\top\bar{g}\right\|_2\right)$$

$$= \mathbb{E}[\mu^\top\mu]\left(\left|(\nabla\log\bar{p}_z V)^\top\nabla\bar{g}\right| + \frac{1}{2}\left\|\nabla\nabla^\top\bar{g}\right\|_2\right). \quad (16)$$

Given the bounding conditions in the theorem, the multiplicative factors to $\mathbb{E}[\mu^\top\mu]$ in the last expressions are bounded by a constant. So when $\frac{1}{\sigma_\mu^2} \to \infty$, *i.e.* $\mathbb{E}[\mu^\top\mu] \to 0$, we have $p(x)$ and $\mathbb{E}[y|x]$ converge uniformly to $(\bar{p}_z V)(x) = f_\#[p_z](x)$ and $\bar{g}(x)$, respectively. So $p(x, y) = p'(x, y)$ indicates $f_\#[p_z] = f'_\#[p'_z]$ and $\bar{g} = \bar{g}'$, which means $\Phi_\#[p_z] = p'_z$ and $p(y|s) = p'(y|\Phi^\mathcal{S}(s, v))$ for any $v \in \mathcal{V}$, due to Eq. (13) and the explanation that follows.

**Proof under condition (iii).** We only need to show that when $\frac{1}{\sigma_\mu^2}$ is much larger than the given quantity, we still have $p(x, y) = p'(x, y) \implies \bar{p}_z V = \bar{p}'_z V', \bar{g} = \bar{g}'$ up to a negligible effect. This task amounts to showing that the residuals $|p(x) - (\bar{p}_z V)(x)|$ and $|\mathbb{E}[y|x] - \bar{g}(x)|$ controlled by Eqs. (15, 16) are negligible. To achieve this, we need to further expand the controlling functions using derivatives of $f$, $g$ and $p_z$ explicitly, and bound them by the bounding constants. In the following, we use indices $a, b, c$ for the components of $x$ and $i, j, k$ for those of $z$. For functions of $z$ appearing in the following (*e.g.*, $f$, $g$, $p_z$ and their derivatives), they are evaluated at $z = f^{-1}(x)$ since we are bounding functions of $x$.

**(1)** Bounding $|\mathbb{E}[y|x] - \bar{g}(x)| \leqslant \mathbb{E}[\mu^\top\mu]\left(\left|(\nabla\log\bar{p}_z V)^\top\nabla\bar{g}\right| + \frac{1}{2}\left\|\nabla\nabla^\top\bar{g}\right\|_2\right)$ from Eq. (16).

From the chain rule of differentiation, it is easy to show that:

$$\nabla\log\bar{p}_z = J_{f^{-1}}\nabla\log p_z, \qquad \nabla\bar{g} = J_{(f^{-1})^\mathcal{S}}\nabla g = J_{f^{-1}}\nabla_z g, \quad (17)$$

where $\nabla_z g = (\nabla g^\top, 0_{d_\nu}^\top)^\top$ (recall that $g$ is a function only of $s$). For the term $\nabla \log V$, we apply Jacobi's formula for the derivative of the log-determinant:

$$\partial_a \log V(x) = \partial_a \log |J_{f^{-1}}(x)| = \operatorname{tr}\left(J_{f^{-1}}^{-1}(x)\big(\partial_a J_{f^{-1}}(x)\big)\right) = \sum_{b,i} J_{f^{-1}}^{-1}(x)_{ib}\big(\partial_a J_{f^{-1}}(x)_{bi}\big)$$

$$= \sum_{b,i} J_f(f^{-1}(x))_{ib}\partial_b\partial_a f_i^{-1}(x) = \sum_i \big(J_f(\nabla\nabla^\top f_i^{-1})\big)_{ia}. \tag{18}$$

However, as bounding Eq. (17) already requires bounding $\|J_{f^{-1}}\|_2$, directly using this expression to bound $\|\nabla \log V\|_2$ would require to also bound $\|J_f\|_2$. This requirement to bound the first-order derivatives of both $f$ and $f^{-1}$ is a relatively restrictive one. To ease the requirement, we would like to express $\nabla \log V$ in terms of $J_{f^{-1}}$. This can be achieved by expressing $\nabla\nabla^\top f_i^{-1}$'s in terms of $\nabla\nabla^\top f_c$'s. To do this, first consider a general invertible-matrix-valued function $A(\alpha)$ on a scalar $\alpha$. We have $0 = \partial_\alpha\big(A(\alpha)^{-1}A(\alpha)\big) = (\partial_\alpha A^{-1})A + A^{-1}\partial_\alpha A$, so we have $A^{-1}\partial_\alpha A = -(\partial_\alpha A^{-1})A$, consequently $\partial_\alpha A = -A(\partial_\alpha A^{-1})A$. Using this relation (in the fourth equality below), we have:

$$\big(\nabla\nabla^\top f_i^{-1}\big)_{ab} = \partial_a\partial_b f_i^{-1} = \partial_a\big(J_{f^{-1}}\big)_{bi} = \big(\partial_a J_{f^{-1}}\big)_{bi}$$

$$= -\Big(J_{f^{-1}}(\partial_a J_{f^{-1}}^{-1})J_{f^{-1}}\Big)_{bi} = -\Big(J_{f^{-1}}\big(\partial_a J_f\big)J_{f^{-1}}\Big)_{bi}$$

$$= -\sum_{jc}(J_{f^{-1}})_{bj}\big(\partial_a(\partial_j f_c)\big)(J_{f^{-1}})_{ci} = -\sum_{jck}(J_{f^{-1}})_{bj}(\partial_k\partial_j f_c)(\partial_a f_k^{-1})(J_{f^{-1}})_{ci}$$

$$= -\sum_c (J_{f^{-1}})_{ci}\sum_{jk}(J_{f^{-1}})_{bj}(\partial_k\partial_j f_c)(J_{f^{-1}})_{ak} = -\sum_c (J_{f^{-1}})_{ci}\big(J_{f^{-1}}(\nabla\nabla^\top f_c)J_{f^{-1}}^\top\big)_{ab},$$

or in matrix form,

$$\nabla\nabla^\top f_i^{-1} = -\sum_c (J_{f^{-1}})_{ci}J_{f^{-1}}(\nabla\nabla^\top f_c)J_{f^{-1}}^\top =: -\sum_c (J_{f^{-1}})_{ci}K^c, \tag{19}$$

where we have defined the matrix $K^c := J_{f^{-1}}(\nabla\nabla^\top f_c)J_{f^{-1}}^\top$ which is symmetric. Substituting with this result, we can transform Eq. (18) into a desired form:

$$\nabla \log V(x) = \sum_i \big(J_f(\nabla\nabla^\top f_i^{-1})\big)_{i:}^\top = -\sum_i \Big(J_f\sum_c (J_{f^{-1}})_{ci}J_{f^{-1}}(\nabla\nabla^\top f_c)J_{f^{-1}}^\top\Big)_{i:}^\top$$

$$= -\sum_i \Big(\sum_c (J_{f^{-1}})_{ci}J_f J_f^{-1}(\nabla\nabla^\top f_c)J_{f^{-1}}^\top\Big)_{i:}^\top = -\sum_{ci}(J_{f^{-1}})_{ci}\Big((\nabla\nabla^\top f_c)J_{f^{-1}}^\top\Big)_{i:}^\top$$

$$= -\sum_c \Big(J_{f^{-1}}(\nabla\nabla^\top f_c)J_{f^{-1}}^\top\Big)_{c:}^\top = -\sum_c (K_{c:}^c)^\top = -\sum_c K_{:c}^c, \tag{20}$$

so its norm can be bounded by:

$$\|\nabla \log V(x)\|_2 = \Big\|\sum_c K_{c:}^c\Big\|_2 = \Big\|\sum_c (J_{f^{-1}})_{c:}(\nabla\nabla^\top f_c)J_{f^{-1}}^\top\Big\|_2$$

$$\leqslant \sum_c \big\|(J_{f^{-1}})_{c:}\big\|_2\big\|\nabla\nabla^\top f_c\big\|_2\big\|J_{f^{-1}}\big\|_2 \leqslant B_f'' B_{f^{-1}}'\sum_c \big\|(J_{f^{-1}})_{c:}\big\|_2$$

$$\leqslant d B_{f^{-1}}'^2 B_f'', \tag{21}$$

where we have used the following result in the last inequality:

$$\sum_c \big\|(J_{f^{-1}})_{c:}\big\|_2 \leqslant d^{1/2}\sqrt{\sum_c \big\|(J_{f^{-1}})_{c:}\big\|_2^2} = d^{1/2}\big\|J_{f^{-1}}\big\|_F \leqslant d\big\|J_{f^{-1}}\big\|_2 \leqslant d B_{f^{-1}}'. \tag{22}$$

Integrating Eq. (17) and Eq. (21), we have:

$$\big|(\nabla\log \bar{p}_z V)^\top \nabla\bar{g}\big| = (J_{f^{-1}}\nabla\log p_z + \nabla\log V)^\top J_{f^{-1}}\nabla_z g$$

$$\leqslant \big(\|J_{f^{-1}}\|_2\|\nabla\log p_z\|_2 + \|\nabla\log V\|_2\big)\|J_{f^{-1}}\|\|\nabla g\|_2$$

$$\leqslant \big(B_{f^{-1}}' B_{\log p}' + d B_{f^{-1}}'^2 B_f''\big)B_{f^{-1}}' B_g'$$

$$= \big(B_{\log p}' + d B_{f^{-1}}' B_f''\big)B_{f^{-1}}'^2 B_g'. \tag{23}$$

For the Hessian of $\bar{g}$, direct calculus gives:

$$\nabla\nabla^\top \bar{g} = J_{(f^{-1})^\mathcal{S}}(\nabla\nabla^\top g)J_{(f^{-1})^\mathcal{S}}^\top + \sum_{i=1}^{d_\mathcal{S}}(\nabla g)_{s_i}(\nabla\nabla^\top f_{s_i}^{-1})$$

$$= J_{f^{-1}}(\nabla_z\nabla_z^\top g)J_{f^{-1}}^\top + \sum_i (\nabla_z g)_i(\nabla\nabla^\top f_i^{-1}).$$

To avoid the requirement of bounding both $\nabla\nabla^\top f_c$'s and $\nabla\nabla^\top f_i^{-1}$'s, we substitute $\nabla\nabla^\top f_i^{-1}$ using Eq. (19):

$$\nabla\nabla^\top \bar{g} = J_{f^{-1}}(\nabla_z\nabla_z^\top g)J_{f^{-1}}^\top - \sum_i (\nabla_z g)_i \sum_c (J_{f^{-1}})_{ci} K^c$$

$$= J_{f^{-1}}(\nabla_z\nabla_z^\top g)J_{f^{-1}}^\top - \sum_c \Big((J_{f^{-1}})_{c,:}(\nabla_z g)\Big)K^c.$$

So its norm can be bounded by:

$$\big\|\nabla\nabla^\top \bar{g}\big\|_2 \leqslant \big\|J_{f^{-1}}\big\|_2^2\big\|\nabla\nabla^\top g\big\|_2 + \sum_c \big|(J_{f^{-1}})_{c:}(\nabla_z g)\big|\big\|K^c\big\|_2$$

$$\leqslant B_{f^{-1}}'^2 B_g'' + \sum_c \big|(J_{f^{-1}})_{c:}(\nabla_z g)\big| B_{f^{-1}}'^2 B_f''$$

$$\leqslant B_{f^{-1}}'^2\Big(B_g'' + B_f'' \sum_c \big\|(J_{f^{-1}})_{c:}\big\|_2\big\|\nabla_z g\big\|_2\Big)$$

$$\leqslant B_{f^{-1}}'^2\Big(B_g'' + B_f'' B_g' \sum_c \big\|(J_{f^{-1}})_{c:}\big\|_2\Big)$$

$$\leqslant B_{f^{-1}}'^2\Big(B_g'' + dB_{f^{-1}}' B_f'' B_g'\Big), \tag{24}$$

where we have used Eq. (22) in the last inequality. Assembling Eq. (23) and Eq. (24) into Eq. (16), we have:

$$\big|\mathbb{E}[y|x] - \bar{g}(x)\big| \leqslant \mathbb{E}[\mu^\top\mu]B_{f^{-1}}'^2\Big(B_{\log p}' B_g' + \frac{1}{2}B_g'' + \frac{3}{2}dB_{f^{-1}}' B_f'' B_g'\Big). \tag{25}$$

So given the condition **(iii)**, this residual can be neglected.

**(2)** Bounding $|p(x) - (\bar{p}_z V)(x)| \leqslant \frac{1}{2}\mathbb{E}[\mu^\top\mu]|\bar{p}_z V|\big(\|\nabla\log\bar{p}_z V\|_2^2 + \big\|\nabla\nabla^\top\log\bar{p}_z\big\|_2 + \big\|\nabla\nabla^\top\log V\big\|_2\big)$ from Eq. (15).

To begin with, for any $x$, $\bar{p}_z(x) = p_z(f^{-1}(x)) \leqslant B_p$, and $V(x) = \big|J_{f^{-1}}(x)\big|$ is the product of absolute eigenvalues of $J_{f^{-1}}(x)$. Since $\big\|J_{f^{-1}}(x)\big\|_2$ is the largest absolute eigenvalue of $J_{f^{-1}}(x)$, so $V(x) \leqslant \big\|J_{f^{-1}}(x)\big\|_2^d \leqslant B_{f^{-1}}'^d$.

For the first norm in the bracket of the r.h.s of Eq. (15), we have:

$$\|\nabla\log\bar{p}_z V\|_2^2 = \|\nabla\log\bar{p}_z\|_2^2 + 2(\nabla\log\bar{p}_z)^\top\nabla\log V + \|\nabla\log V\|_2^2$$

$$\leqslant \|\nabla\log\bar{p}_z\|_2^2 + 2\|\nabla\log\bar{p}_z\|_2\|\nabla\log V\|_2 + \|\nabla\log V\|_2$$

$$\leqslant B_{f^{-1}}'^2 B_{\log p}'^2 + 2dB_{f^{-1}}'^3 B_f'' B_{\log p}' + \|\nabla\log V\|_2^2, \tag{26}$$

where we have utilized Eq. (17) and Eq. (21) in the last inequality. We consider bounding $\|\nabla\log V\|_2^2$ separately. Using Eq. (20) (in the second equality below), we have:

$$\|\nabla\log V\|_2^2 = \big|(\nabla\log V)^\top(\nabla\log V)\big| = \Big|\sum_c (K_{:c}^c)^\top \sum_d K_{:d}^d\Big|$$

$$= \Big|\sum_{cd} K_{c:}^c K_{:d}^d\Big| \leqslant \sum_{cd}\big|K_{c:}^c K_{:d}^d\big|$$

$$= \sum_{cd}\Big|(J_{f^{-1}})_{c:}(\nabla\nabla^\top f_c)J_{f^{-1}}^\top J_{f^{-1}}(\nabla\nabla^\top f_d)(J_{f^{-1}})_{d:}^\top\Big|$$

$$\leqslant \sum_{cd}\left|(J_{f-1})_{c:}(J_{f-1})_{d:}^{\top}\right|\left\|(\nabla\nabla^{\top}f_c)J_{f-1}^{\top}J_{f-1}(\nabla\nabla^{\top}f_d)\right\|_2$$

$$\leqslant \sum_{cd}\left|(J_{f-1})_{c:}(J_{f-1})_{d:}^{\top}\right|B_{f-1}'^2 B_f''^2 = B_{f-1}'^2 B_f''^2\sum_{cd}\left|(J_{f-1}J_{f-1}^{\top})_{cd}\right|$$

$$\leqslant d^{3/2}B_{f-1}'^2 B_f''^2\left\|J_{f-1}J_{f-1}^{\top}\right\|_2 \leqslant d^{3/2}B_{f-1}'^4 B_f''^2, \tag{27}$$

where we have used the facts for general matrix $A$ and (column) vectors $\alpha, \beta$ that

$$\left|\alpha^{\top}A\beta\right| = \left\|\alpha(A\beta)^{\top}\right\|_2 = \left\|\alpha\beta^{\top}A^{\top}\right\|_2 \leqslant \left\|\alpha\beta^{\top}\right\|_2\|A\|_2 = \left|\alpha^{\top}\beta\right|\|A\|_2 \tag{28}$$

in the fifth last inequality, and that

$$\sum_{cd}|A_{cd}| \leqslant \sqrt{d^2}\sqrt{\sum_{cd}|A_{cd}|^2} = d\|A\|_F \leqslant d^{3/2}\|A\|_2 \tag{29}$$

in the second last inequality. Substituting Eq. (27) into Eq. (26), we have:

$$\|\nabla\log\bar{p}_z V\|_2^2 \leqslant B_{f-1}'^2 B_{\log p}'^2 + 2dB_{f-1}'^3 B_f'' B_{\log p}' + d^{3/2}B_{f-1}'^4 B_f''^2. \tag{30}$$

For the second norm in the bracket of the r.h.s of Eq. (15), similar to Eq. (24), we have:

$$\left\|\nabla\nabla^{\top}\log\bar{p}_z\right\|_2 \leqslant B_{f-1}'^2\left(B_{\log p}'' + dB_{f-1}' B_f'' B_{\log p}'\right). \tag{31}$$

The third norm $\left\|\nabla\nabla^{\top}\log V\right\|_2$ in the bracket of the r.h.s of Eq. (15) needs some more effort. From Eq. (20), we have $\partial_b\log V = -\sum_{cij}(J_{f-1})_{ci}(\partial_i\partial_j f_c)(J_{f-1})_{bj}$, thus

$$\partial_a\partial_b\log V = -\sum_{cij}\partial_a(J_{f-1})_{ci}(\partial_i\partial_j f_c)(J_{f-1})_{bj} - \sum_{cij}(J_{f-1})_{ci}(\partial_i\partial_j f_c)\partial_a(J_{f-1})_{bj}$$

$$-\sum_{cij}(J_{f-1})_{ci}\partial_a(\partial_i\partial_j f_c)(J_{f-1})_{bj}$$

$$= -\sum_{cij}(\partial_a\partial_c f_i^{-1})(\partial_i\partial_j f_c)(J_{f-1})_{bj} - \sum_{cij}(J_{f-1})_{ci}(\partial_i\partial_j f_c)(\partial_a\partial_b f_j^{-1})$$

$$-\sum_{cijk}(J_{f-1})_{ci}(\partial_a f_k^{-1})(\partial_k\partial_i\partial_j f_c)(J_{f-1})_{bj}$$

$$= \sum_{cijd}(J_{f-1})_{di}K_{ac}^d(\partial_i\partial_j f_c)(J_{-1})_{bj} + \sum_{cijd}(J_{f-1})_{ci}(\partial_i\partial_j f_c)(J_{f-1})_{dj}K_{ab}^d$$

$$-\sum_{cijk}(J_{f-1})_{ci}(\partial_k\partial_i\partial_j f_c)(J_{f-1})_{ak}(J_{f-1})_{bj}$$

$$= \sum_{cd}K_{ac}^d K_{db}^c + \sum_{cd}K_{cd}^c K_{ab}^d - \sum_{cijk}(J_{f-1})_{ci}(\partial_k\partial_i\partial_j f_c)(J_{f-1})_{ak}(J_{f-1})_{bj},$$

where we have used Eq. (19) in the third equality for the first two terms. In matrix form, we have:

$$\nabla\nabla^{\top}\log V = \sum_{cd}K_{:c}^d K_{d:}^c + \sum_{cd}K_{cd}^c K^d - \sum_{cijk}(J_{f-1})_{ci}(\partial_k\partial_i\partial_j f_c)(J_{f-1})_{:k}(J_{f-1})_{:j}^{\top}.$$

We now bound the norms of the three terms in turn. For the first term,

$$\left\|\sum_{cd}K_{:c}^d K_{d:}^c\right\|_2 \leqslant \sum_{cd}\left\|K_{:c}^d K_{d:}^c\right\|_2 = \sum_{cd}\left|K_{d:}^c K_{:c}^d\right|$$

$$= \sum_{cd}\left|(J_{f-1})_{d:}(\nabla\nabla^{\top}f_c)J_{f-1}^{\top}J_{f-1}(\nabla\nabla^{\top}f_d)(J_{f-1})_{c:}^{\top}\right|$$

$$\leqslant \sum_{cd}\left|(J_{f-1})_{d:}(J_{f-1})_{c:}^{\top}\right|\left\|(\nabla\nabla^{\top}f_c)J_{f-1}^{\top}J_{f-1}(\nabla\nabla^{\top}f_d)\right\|_2$$

$$\leqslant B_{f-1}'^2 B_f''^2\sum_{cd}\left|(J_{f-1}J_{f-1}^{\top})_{dc}\right| \leqslant d^{3/2}B_{f-1}'^2 B_f''^2\left\|J_{f-1}J_{f-1}^{\top}\right\|_2$$

$$\leqslant d^{3/2} B_{f-1}'^4 B_f''^2, \tag{32}$$

where we have used Eq. (28) in the fourth last inequality and Eq. (29) in the second last inequality. For the second term,

$$\left\| \sum_{cd} K_{cd}^c K^d \right\|_2 \leqslant \sum_{cd} |K_{cd}^c| \|K^d\|_2 \leqslant B_{f-1}'^2 B_f'' \sum_{cd} |K_{cd}^c|$$

$$\leqslant d^{1/2} B_{f-1}'^2 B_f'' \sum_c \sqrt{\sum_d |K_{cd}^c|^2} = d^{1/2} B_{f-1}'^2 B_f'' \sum_c \|K_{c:}^c\|_2$$

$$\leqslant d^{1/2} B_{f-1}'^2 B_f'' \sum_c \left\|(J_{f-1})_{c:}\right\|_2 \left\|(\nabla \nabla^\top f_c) J_{f-1}^\top\right\|_2 \leqslant d^{1/2} B_{f-1}'^3 B_f''^2 \sum_c \left\|(J_{f-1})_{c:}\right\|_2$$

$$\leqslant d^{3/2} B_{f-1}'^4 B_f''^2, \tag{33}$$

where we have used Eq. (22) in the last inequality. For the third term,

$$\left\| \sum_{cijk} (J_{f-1})_{ci} (\partial_k \partial_i \partial_j f_c) (J_{f-1})_{:k} (J_{f-1})_{:j}^\top \right\|_2$$

$$\leqslant \sum_{cijk} |(J_{f-1})_{ci} (\partial_k \partial_i \partial_j f_c)| \left\|(J_{f-1})_{:k} (J_{f-1})_{:j}^\top\right\|_2 \leqslant B_f''' \sum_{ci} |(J_{f-1})_{ci}| \sum_{jk} \left\|(J_{f-1})_{:k} (J_{f-1})_{:j}^\top\right\|_2$$

$$\leqslant d^{3/2} B_f''' \|J_{f-1}\|_2 \sum_{jk} |(J_{f-1})_{:k}^\top (J_{f-1})_{:j}| \leqslant d^{3/2} B_f''' B_{f-1}' \sum_{jk} \left|(J_{f-1}^\top J_{f-1})_{kj}\right|$$

$$\leqslant d^3 B_f''' B_{f-1}' \left\|J_{f-1}^\top J_{f-1}\right\|_2 \leqslant d^3 B_f''' B_{f-1}'^3, \tag{34}$$

where we have used Eq. (29) in the fourth last and second last inequalities.

Finally, by assembling Eqs. (30, 31, 32, 33, 34) into Eq. (15), we have:

$$|p(x) - (\bar{p}_z V)(x)| \leqslant \frac{1}{2} \mathbb{E}[\mu^\top \mu] B_p B_{f-1}'^d \big( B_{f-1}'^2 B_{\log p}'^2 + 2d B_{f-1}'^3 B_f'' B_{\log p}' + d^{3/2} B_{f-1}'^4 B_f''^2$$

$$+ B_{f-1}'^2 (B_{\log p}'' + d B_{f-1}' B_f'' B_{\log p}') + 2 d^{3/2} B_{f-1}'^4 B_f''^2 + d^3 B_f''' B_{f-1}'^3 \big)$$

$$= \frac{1}{2} \mathbb{E}[\mu^\top \mu] B_p B_{f-1}'^{d+2} \big( B_{\log p}'^2 + B_{\log p}'' + 3d B_{f-1}' B_f'' B_{\log p}'$$

$$+ 3 d^{3/2} B_{f-1}'^2 B_f''^2 + d^3 B_f''' B_{f-1}' \big).$$

So given the condition (iii), this residual can be neglected. $\qquad \square$

## A.3 PROOF OF THE OOD GENERALIZATION ERROR BOUND THEOREM 5.5

We give the following more detailed version of Theorem 5.5 and prove it. The theorem in the main context corresponds to conclusion (ii) below (*i.e.*, Eq. (37) below recovers Eq. (6)) by taking the CSGs $p'$, $p$ and $\tilde{p}$ as the semantic-identified CSG $p$ on the training domain and the ground-truth CSGs on the training $p^*$ and test $\tilde{p}^*$ domains, respectively. Here, the semantic-identification requirement on the learned CSG $p$ is to guarantee that it is semantic-equivalent to the ground-truth CSG $p^*$ on the training domain, so that the condition in (ii) is satisfied.

**Theorem 5.5'** (OOD generalization error). *Let Assumptions 5.1 and 5.2 hold.* (i) *Consider two CSGs $p$ and $\tilde{p}$ that share the same generative mechanisms $p(x|s,v)$ and $p(y|s)$ but have different priors $p_{s,v}$ and $\tilde{p}_{s,v}$. Then up to $O(\sigma_\mu^2)$ where $\sigma_\mu^2 := \mathbb{E}[\mu^\top \mu]$, we have for any $x \in \text{supp}(p_x) \cap \text{supp}(\tilde{p}_x)$,*

$$\left| \mathbb{E}[y|x] - \tilde{\mathbb{E}}[y|x] \right| \leqslant \sigma_\mu^2 \|\nabla g\|_2 \|J_{f-1}\|_2^2 \|\nabla \log(p_{s,v}/\tilde{p}_{s,v})\|_2 \Big|_{(s,v)=f^{-1}(x)}, \tag{35}$$

*where $J_{f-1}$ is the Jacobian of $f^{-1}$. Further assume that the bounds $B$'s defined in* Theorem 5.4'(iii) *hold. Then the error is negligible for any $x \in \text{supp}(p_x) \cap \text{supp}(\tilde{p}_x)$ if $\frac{1}{\sigma_\mu^2} \gg B_{\log p}' B_g' B_{f-1}'^2$, and:*

$$\mathbb{E}_{\tilde{p}(x)} \left| \mathbb{E}[y|x] - \tilde{\mathbb{E}}[y|x] \right|^2 \leqslant \sigma_\mu^4 B_g'^2 B_{f-1}'^4 \mathbb{E}_{\tilde{p}_{s,v}} [2\Delta \log p_{s,v} - \Delta \log \tilde{p}_{s,v} + \|\nabla \log p_{s,v}\|_2^2] \tag{36}$$

*if $\text{supp}(p_x) = \text{supp}(\tilde{p}_x)$, where $\Delta$ denotes the Laplacian operator.*

**(ii)** *Let $p'$ be a CSG that is semantic-equivalent to the CSG $p$ introduced in* **(i)**. *Then up to $O(\sigma_\mu^2)$, we have for any $x \in \mathrm{supp}(p'_x) \cap \mathrm{supp}(\tilde{p}_x)$,*

$$\left|\mathbb{E}'[y|x] - \tilde{\mathbb{E}}[y|x]\right| \leqslant \sigma_\mu^2 \|\nabla g'\|_2 \big\|J_{f'^{-1}}\big\|_2^2 \big\|\nabla \log(p'_{s,v}/\tilde{p}'_{s,v})\big\|_2 \bigg|_{(s,v)=f'^{-1}(x)}, \tag{37}$$

*where $\tilde{p}'_{s,v} := \Phi_\#[\tilde{p}_{s,v}]$ is the prior of CSG $\tilde{p}$ under the parameterization of CSG $p'$, derived as the pushed-forward distribution by the reparameterization $\Phi := f'^{-1} \circ f$ from $p$ to $p'$.*

For conclusion **(i)**, in the expected OOD generalization error in Eq. (36), the term $\mathbb{E}_{\tilde{p}_{s,v}}[2\Delta \log p_{s,v} - \Delta \log \tilde{p}_{s,v} + \|\nabla \log p_{s,v}\|_2^2]$ is actually the score matching objective (Fisher divergence) (Hyvärinen, 2005) that measures the difference between $\tilde{p}_{s,v}$ and $p_{s,v}$. For Gaussian priors $p(s,v) = \mathcal{N}(0, \Sigma)$ and $\tilde{p}(s,v) = \mathcal{N}(0, \tilde{\Sigma})$, the term reduces to the matrix trace, $\mathrm{tr}(-2\Sigma^{-1} + \tilde{\Sigma}^{-1} + \Sigma^{-1}\tilde{\Sigma}\Sigma^{-1})$. For $\Sigma = \tilde{\Sigma}$, the term vanishes.

For conclusion **(ii)**, note that since $p$ and $p'$ are semantic-equivalent, we have $p'_x = p_x$ and $\mathbb{E}'[y|x] = \mathbb{E}[y|x]$ (from Lemma A.2). So Eqs. (35, 37) bound the same quantity. Equation (37) expresses the bound using the structures of the CSG $p'$. It is considered since recovering the exact CSG $p$ from $(x, y)$ data is impractical and we can only learn a CSG $p'$ that is semantic-equivalent to $p$.

*Proof.* Following the proof A.2 of Theorem 5.4', we assume the additive noise variables $\mu$ and $\nu$ (for continuous $y$) have zero mean without loss of generality, and we denote $z := (s, v)$.

**Proof under condition (i).** Under the assumptions, we have Eq. (14) in the proof A.2 of Theorem 5.4' hold. Noting that the two CSGs share the same $\bar{g}$ and $V$ (since they share the same $p(x|s,v)$ and $p(y|s)$ thus $f$ and $g$), we have for any $x \in \mathrm{supp}(p_x) \cap \mathrm{supp}(\tilde{p}_x)$,

$$\mathbb{E}[y|x] = \bar{g} + \frac{1}{2}\mathbb{E}_{p(\mu)}\left[\mu^\top\big((\nabla \log \bar{p}_z V)\nabla \bar{g}^\top + \nabla \bar{g}(\nabla \log \bar{p}_z V)^\top + \nabla\nabla^\top \bar{g})\mu\right] + O(\sigma_\mu^3),$$

$$\tilde{\mathbb{E}}[y|x] = \bar{g} + \frac{1}{2}\mathbb{E}_{p(\mu)}\left[\mu^\top\big((\nabla \log \tilde{\bar{p}}_z V)\nabla \bar{g}^\top + \nabla \bar{g}(\nabla \log \tilde{\bar{p}}_z V)^\top + \nabla\nabla^\top \bar{g})\mu\right] + O(\sigma_\mu^3), \tag{38}$$

where we have similarly defined $\tilde{\bar{p}}_z := \tilde{p}_z \circ f^{-1}$. By subtracting the two equations, we have that up to $O(\sigma_\mu^2)$,

$$\left|\mathbb{E}[y|x] - \tilde{\mathbb{E}}[y|x]\right| = \frac{1}{2}\left|\mathbb{E}_{p(\mu)}\left[\mu^\top\big(\nabla \log(\bar{p}_z/\tilde{\bar{p}}_z)\nabla \bar{g}^\top + \nabla \bar{g}\nabla \log(\bar{p}_z/\tilde{\bar{p}}_z)^\top\big)\mu\right]\right|$$

$$\leqslant \frac{1}{2}\mathbb{E}_{p(\mu)}\left[\left|\mu^\top\big(\nabla \log(\bar{p}_z/\tilde{\bar{p}}_z)\nabla \bar{g}^\top + \nabla \bar{g}\nabla \log(\bar{p}_z/\tilde{\bar{p}}_z)^\top\big)\mu\right|\right]$$

$$\leqslant \frac{1}{2}\mathbb{E}_{p(\mu)}\left[\|\mu\|_2^2\big(\big\|\nabla \log(\bar{p}_z/\tilde{\bar{p}}_z)\nabla \bar{g}^\top\big\|_2 + \big\|\nabla \bar{g}\nabla \log(\bar{p}_z/\tilde{\bar{p}}_z)^\top\big\|_2\big)\right]$$

$$= \left|\nabla \bar{g}^\top \nabla \log(\bar{p}_z/\tilde{\bar{p}}_z)\right|\mathbb{E}[\mu^\top \mu]. \tag{39}$$

The multiplicative factor to $\mathbb{E}[\mu^\top \mu]$ on the right hand side can be further bounded by:

$$\left|\nabla \bar{g}^\top \nabla \log(\bar{p}_z/\tilde{\bar{p}}_z)\right| = \left|(J_{(f^{-1})^S}\nabla g)^\top (J_{f^{-1}}\nabla \log(p_z/\tilde{p}_z))\right|$$

$$= \left|\nabla g^\top J_{(f^{-1})^S}^\top J_{f^{-1}}\nabla \log(p_z/\tilde{p}_z)\right|$$

$$= \left|((\nabla g)^\top, 0_{d_V}^\top)J_{f^{-1}}^\top J_{f^{-1}}\nabla \log(p_z/\tilde{p}_z)\right|$$

$$\leqslant \|\nabla g\|_2 \big\|J_{f^{-1}}\big\|_2^2 \|\nabla \log(p_z/\tilde{p}_z)\|_2, \tag{40}$$

where $\nabla g$ and $\nabla \log(p_z/\tilde{p}_z)$ are evaluated at $z = f^{-1}(x)$. This gives:

$$\left|\mathbb{E}[y|x] - \tilde{\mathbb{E}}[y|x]\right| \leqslant \sigma_\mu^2 \|\nabla g\|_2 \big\|J_{f^{-1}}\big\|_2^2 \|\nabla \log(p_z/\tilde{p}_z)\|_2,$$

*i.e.* Eq. (35) in conclusion **(i)**. When the bounds $B$'s in Theorem 5.4'**(iii)** hold, we further have:

$$\left|\mathbb{E}[y|x] - \tilde{\mathbb{E}}[y|x]\right| \leqslant \sigma_\mu^2 \|\nabla g\|_2 \big\|J_{f^{-1}}\big\|_2^2 \|\nabla \log p_z - \nabla \log \tilde{p}_z\|_2$$

$$\leqslant \sigma_\mu^2 \|\nabla g\|_2 \big\|J_{f^{-1}}\big\|_2^2 (\|\nabla \log p_z\|_2 + \|\nabla \log \tilde{p}_z\|_2)$$

$$\leqslant 2\sigma_\mu^2 B_g' B_{f^{-1}}'^2 B_{\log p}'.$$

So when $\frac{1}{\sigma_\mu^2} \gg B_{\log p}' B_g' B_{f^{-1}}'^2$, this difference is negligible for any $x \in \mathrm{supp}(p_x) \cap \mathrm{supp}(\tilde{p}_x)$.

We now turn to the expected OOD generalization error Eq. (36) in conclusion **(i)**. When $\mathrm{supp}(p_x) = \mathrm{supp}(\tilde{p}_x)$, Eq. (35) hold on $\tilde{p}_x$. Together with the bounds in Theorem 5.4'**(iii)**, we have:

$$\mathbb{E}_{\tilde{p}(x)}\Big|\mathbb{E}[y|x] - \tilde{\mathbb{E}}[y|x]\Big|^2 \leqslant \sigma_\mu^4 B_g'^2 B_{f^{-1}}'^4 \mathbb{E}_{\tilde{p}(x)}\Big\|\nabla \log(p_z/\tilde{p}_z)\big|_{z=f^{-1}(x)}\Big\|_2^2$$
$$= \sigma_\mu^4 B_g'^2 B_{f^{-1}}'^4 \mathbb{E}_{\tilde{p}_z}\|\nabla \log(p_z/\tilde{p}_z)\|_2^2,$$

where the equality holds due to the generating process of the model. Note that the term $\mathbb{E}_{\tilde{p}_z}\|\nabla \log(p_z/\tilde{p}_z)\|_2^2$ therein is the score matching objective (Fisher divergence). By Hyvärinen (2005, Theorem 1), we can reformulate it as $\mathbb{E}_{\tilde{p}_z}[2\Delta \log p_z - \Delta \log \tilde{p}_z + \|\nabla \log p_z\|_2^2]$, so we have:

$$\mathbb{E}_{\tilde{p}(x)}\Big|\mathbb{E}[y|x] - \tilde{\mathbb{E}}[y|x]\Big|^2 \leqslant \sigma_\mu^4 B_g'^2 B_{f^{-1}}'^4 \mathbb{E}_{\tilde{p}_z}[2\Delta \log p_z - \Delta \log \tilde{p}_z + \|\nabla \log p_z\|_2^2].$$

**Proof under condition (ii).** From Eq. (14) in the proof A.2 of Theorem 5.4', we have for CSG $p'$ that for any $x \in \mathrm{supp}(p_x')$ or equivalently $x \in \mathrm{supp}(p_x)$,

$$\mathbb{E}'[y|x] = \bar{g}' + \frac{1}{2}\mathbb{E}_{p(\mu)}\big[\mu^\top((\nabla \log \bar{p}_z' V')\nabla \bar{g}'^\top + \nabla \bar{g}'(\nabla \log \bar{p}_z' V')^\top + \nabla\nabla^\top \bar{g}')\mu\big] + O(\sigma_\mu^3), \quad (41)$$

where we have similarly defined $\bar{p}_z' := p_z' \circ f'^{-1}$ and $\bar{g}' := g' \circ (f'^{-1})^{\mathcal{S}}$. Since $p$ and $p'$ are semantic-equivalent with reparameterization $\Phi$ from $p$ to $p'$, we have $p(y|s) = p'(y|\Phi^{\mathcal{S}}(s,v))$ thus $g(s) = g'(\Phi^{\mathcal{S}}(s,v))$ for any $v \in \mathcal{V}$. So for any $x \in \mathrm{supp}(p_x)$ or equivalently $x \in \mathrm{supp}(p_x')$, we have $g((f^{-1})^{\mathcal{S}}(x)) = g'(\Phi^{\mathcal{S}}((f^{-1})^{\mathcal{S}}(x), (f^{-1})^{\mathcal{V}}(x))) = g'(\Phi^{\mathcal{S}}(f^{-1}(x))) = g'((f'^{-1})^{\mathcal{S}}(f(f^{-1}(x)))) = g'((f'^{-1})^{\mathcal{S}}(x))$, i.e., $\bar{g} = \bar{g}'$. For another fact, since $\tilde{p}_z' := \Phi_\#[\tilde{p}_z] = (f'^{-1}\circ f)_\#[\tilde{p}_z]$ by definition, we have $f'_\#[\tilde{p}_z'] = f_\#[\tilde{p}_z]$, i.e., $\bar{\tilde{p}}_z' V' = \bar{\tilde{p}}_z V$. Subtracting Eqs. (41, 38) and applying these two facts, we have up to $O(\sigma_\mu^2)$, for any $x \in \mathrm{supp}(p_x') \cap \mathrm{supp}(\tilde{p}_x)$,

$$\Big|\mathbb{E}'[y|x] - \tilde{\mathbb{E}}[y|x]\Big| = \frac{1}{2}\Big|\mathbb{E}_{p(\mu)}\big[\mu^\top\big(\nabla \log(\bar{p}_z'/\bar{\tilde{p}}_z')\nabla \bar{g}'^\top + \nabla \bar{g}'\nabla \log(\bar{p}_z'/\bar{\tilde{p}}_z')^\top\big)\mu\big]\Big|$$
$$\leqslant \Big|\nabla \bar{g}'^\top \nabla \log(\bar{p}_z'/\bar{\tilde{p}}_z')\Big|\mathbb{E}[\mu^\top \mu],$$

where the inequality follows Eq. (39). Using a similar result of Eq. (40), we have:

$$\Big|\mathbb{E}'[y|x] - \tilde{\mathbb{E}}[y|x]\Big| \leqslant \sigma_\mu^2\|\nabla g'\|_2\big\|J_{f'^{-1}}\big\|_2^2\big\|\nabla \log(p_z'/\tilde{p}_z')\big\|_2,$$

where $\nabla g'$ and $\nabla \log(p_z'/\tilde{p}_z')$ are evaluated at $z = f'^{-1}(x)$. This gives Eq. (37). $\qquad\square$

## A.4 PROOF OF THE DOMAIN ADAPTATION ERROR THEOREM 5.6

To be consistent with the notation in the proofs, we prove the theorem by denoting the semantic-identified CSG $p$ and the ground-truth CSG $\tilde{p}^*$ on the test domain as $p'$ and $\tilde{p}$, respectively.

*Proof.* The new prior $\tilde{p}'(z)$ is learned by fitting unsupervised data from the test domain $\tilde{p}(x)$. Applying the deduction in the proof A.2 of Theorem 5.4' to the test domain, we have that under any of the three conditions in Theorem 5.4', $\tilde{p}(x) = \tilde{p}'(x)$ indicates $f_\#[\tilde{p}_z] = f'_\#[\tilde{p}_z']$. This gives $\tilde{p}_z' = (f'^{-1}\circ f)_\#[\tilde{p}_z] = \Phi_\#[\tilde{p}_z]$.

From Eq. (12) in the same proof, we have that:

$$\tilde{p}(x)\tilde{\mathbb{E}}[y|x] = (f_\#[g\tilde{p}_z] * p_\mu)(x) = ((f_\#[\tilde{p}_z]\bar{g}) * p_\mu)(x),$$
$$\tilde{p}'(x)\tilde{\mathbb{E}}'[y|x] = (f'_\#[g'\tilde{p}_z'] * p_\mu)(x) = ((f'_\#[\tilde{p}_z']\bar{g}') * p_\mu)(x).$$

From the proof A.3 of Theorem 5.5'**(ii)** (the paragraph under Eq. (41)), the semantic-equivalence between CSGs $p$ and $p'$ indicates that $\bar{g} = \bar{g}'$. So from the above two equations, we have $\tilde{p}(x)\tilde{\mathbb{E}}[y|x] = \tilde{p}'(x)\tilde{\mathbb{E}}'[y|x]$ (recall that $\tilde{p}(x) = \tilde{p}'(x)$ indicates $f_\#[\tilde{p}_z] = f'_\#[\tilde{p}_z']$). Since $\tilde{p}(x) = \tilde{p}'(x)$ (that is how

$\tilde{p}'_z$ is learned), we have for any $x \in \mathrm{supp}(\tilde{p}_x)$ or equivalently $x \in \mathrm{supp}(\tilde{p}'_x)$,

$$\tilde{\mathbb{E}}'[y|x] = \tilde{\mathbb{E}}[y|x]. \tag{42}$$

□

# B  ALTERNATIVE IDENTIFIABILITY THEORY FOR CSG

The presented identifiability theory, particularly Theorem 5.4, shows that the semantic-identifiability can be achieved in the deterministic limit ($\frac{1}{\sigma_\mu^2} \to \infty$), but does not quantitatively describe the extent of violation of the identifiability for a finite variance $\sigma_\mu^2$. Here we define a "soft" version of semantic-equivalence and show that it can be achieved with a finite variance, with a trade-off between the "softness" and the variance.

**Definition B.1** ($\delta$-semantic-dependency)**.** For $\delta > 0$ and two CSGs $p$ and $p'$, we say that they are $\delta$-semantic-dependent, if there exists a homeomorphism $\Phi$ on $\mathcal{S} \times \mathcal{V}$ such that: **(i)** $p(x|s,v) = p'(x|\Phi(s,v))$, **(ii)** $\sup_{v \in \mathcal{V}} \left\| g(s) - g'(\Phi^{\mathcal{S}}(s,v)) \right\|_2 \leqslant \delta$ where we have denoted $g(s) := \mathbb{E}[y|s]$, and **(iii)** $\sup_{v^{(1)}, v^{(2)} \in \mathcal{V}} \left\| \Phi^{\mathcal{S}}(s, v^{(1)}) - \Phi^{\mathcal{S}}(s, v^{(2)}) \right\|_2 \leqslant \delta$.

In the definition, we have released the prior conversion requirement, and relaxed the exact likelihood conversion for $p(y|s)$ in **(ii)** and the $v$-constancy of $\Phi^{\mathcal{S}}$ in **(iii)** to allow an error bounded by $\delta$. When $\delta = 0$, the $v$-constancy of $\Phi^{\mathcal{S}}$ is exact, and under the additive noise Assumption 5.1 we also have the exact likelihood conversion $p(y|s) = p'(y|\Phi^{\mathcal{S}}(s,v))$ for any $v \in \mathcal{V}$. So 0-semantic-dependency with the prior conversion requirement reduces to the semantic-equivalence.

Due to the quantitative nature, the binary relation cannot be made an equivalence relation but only a dependency. Here, a dependency refers to a binary relation with reflexivity and symmetry, but no transitivity.

**Proposition B.2.** *The $\delta$-semantic-dependency is a dependency relation if the function $g := \mathbb{E}[y|s]$ is bijective and its inverse $g^{-1}$ is $\frac{1}{2}$-Lipschitz.*

*Proof.* Showing a dependency relation amounts to showing the following two properties.

- Reflexivity. For two identical CSGs $p$ and $p'$, we have $p(x|s,v) = p'(x|s,v)$ and $p(y|s) = p'(y|s)$. So the identity map as $\Phi$ obviously satisfies all the requirements in Definition B.1.
- Symmetry. Let CSG $p$ be $\delta$-semantic-dependent to CSG $p'$ with homeomorphism $\Phi$. Obviously $\Phi^{-1}$ is also a homeomorphism. For any $(s', v') \in \mathcal{S} \times \mathcal{V}$, we have $p'(x|s',v') = p'(x|\Phi(\Phi^{-1}(s',v'))) = p(x|\Phi^{-1}(s',v'))$, and $\left\| g'(s') - g((\Phi^{-1})^{\mathcal{S}}(s',v')) \right\|_2 = \left\| g'(\Phi^{\mathcal{S}}(s,v)) - g(s) \right\|_2 \leqslant \delta$ where we have denoted $(s,v) := \Phi^{-1}(s',v')$ here. So $\Phi^{-1}$ satisfies requirements **(i)** and **(ii)** in Definition B.1.

  For requirement **(iii)**, we need the following fact: for any $s^{(1)}, s^{(2)} \in \mathcal{S}$, $\left\| s^{(1)} - s^{(2)} \right\|_2 = \left\| g^{-1}(g(s^{(1)})) - g^{-1}(g(s^{(2)})) \right\|_2 \leqslant \frac{1}{2} \left\| g(s^{(1)}) - g(s^{(2)}) \right\|_2$, where the inequality holds since $g^{-1}$ is $\frac{1}{2}$-Lipschitz. Then for any $s' \in \mathcal{S}$, we have:

$$\sup_{v'^{(1)}, v'^{(2)} \in \mathcal{V}} \left\| (\Phi^{-1})^{\mathcal{S}}(s', v'^{(1)}) - (\Phi^{-1})^{\mathcal{S}}(s', v'^{(2)}) \right\|_2$$

$$\leqslant \sup_{v'^{(1)}, v'^{(2)} \in \mathcal{V}} \frac{1}{2} \left\| g\big((\Phi^{-1})^{\mathcal{S}}(s', v'^{(1)})\big) - g\big((\Phi^{-1})^{\mathcal{S}}(s', v'^{(2)})\big) \right\|_2$$

$$= \sup_{v'^{(1)}, v'^{(2)} \in \mathcal{V}} \frac{1}{2} \left\| \Big( g\big((\Phi^{-1})^{\mathcal{S}}(s', v'^{(1)})\big) - g'(s') \Big) - \Big( g\big((\Phi^{-1})^{\mathcal{S}}(s', v'^{(2)})\big) - g'(s') \Big) \right\|_2$$

$$\leqslant \sup_{v'^{(1)}, v'^{(2)} \in \mathcal{V}} \frac{1}{2} \left( \left\| g\big((\Phi^{-1})^{\mathcal{S}}(s', v'^{(1)})\big) - g'(s') \right\|_2 + \left\| g\big((\Phi^{-1})^{\mathcal{S}}(s', v'^{(2)})\big) - g'(s') \right\|_2 \right)$$

$$= \frac{1}{2} \left( \sup_{v'^{(1)} \in \mathcal{V}} \left\| g\big((\Phi^{-1})^{\mathcal{S}}(s', v'^{(1)})\big) - g'(s') \right\|_2 + \sup_{v'^{(2)} \in \mathcal{V}} \left\| g\big((\Phi^{-1})^{\mathcal{S}}(s', v'^{(2)})\big) - g'(s') \right\|_2 \right)$$

$$\leqslant \delta,$$

where in the last inequality we have used the fact that $\Phi^{-1}$ satisfies requirement **(ii)**. So $p'$ is $\delta$-semantic-dependent to $p$ via the homeomorphism $\Phi^{-1}$.

$\square$

The corresponding $\delta$-semantic-identifiability result follows.

**Theorem B.3** ($\delta$-semantic-identifiability)**.** *Assume the same as Theorem 5.4' and Proposition B.2, and let the bounds $B$'s defined in* Theorem 5.4'**(iii)** *hold. For two such CSGs $p$ and $p'$, if they have $p(x, y) = p'(x, y)$, then they are $\delta$-semantic-dependent for any $\delta \geqslant \sigma_\mu^2 B_{f^{-1}}'^2 \big(2B_{\log p}' B_g' + B_g'' + 3dB_{f^{-1}}' B_f'' B_g'\big)$, where $d := d_{\mathcal{S}} + d_{\mathcal{V}}$.*

*Proof.* Let $\Phi := f'^{-1} \circ f$, where $f$ and $f'$ are given by the two CSGs $p$ and $p'$ via Assumption 5.1. We now show that $p$ and $p'$ are $\delta$-semantic-dependent via this $\Phi$ for any $\delta$ in the theorem. Obviously $\Phi$ is a homeomorphism on $\mathcal{S} \times \mathcal{V}$, and it satisfies requirement **(i)** in Definition B.1 by construction due to Eq. (7) in the proof A.2 of Theorem 5.4'.

Consider requirement **(ii)** in Definition B.1. Based on the same assumptions as Theorem 5.4', we have Eq. (25) hold for both CSGs:

$$\max\{\|\mathbb{E}[y|x] - \bar{g}(x)\|_2, \|\mathbb{E}'[y|x] - \bar{g}'(x)\|_2\} \leqslant \sigma_\mu^2 B_{f^{-1}}'^2 \big(B_{\log p}' B_g' + \frac{1}{2}B_g'' + \frac{3}{2}dB_{f^{-1}}' B_f'' B_g'\big),$$

where we have denoted $\sigma_\mu^2 := \mathbb{E}[\mu^\top \mu]$. Since both CSGs induce the same $p(y|x)$, so $\mathbb{E}[y|x] = \mathbb{E}'[y|x]$. This gives:

$$\|\bar{g}(x) - \bar{g}'(x)\|_2 = \big\|\big(\mathbb{E}'[y|x] - \bar{g}'(x)\big) - \big(\mathbb{E}[y|x] - \bar{g}(x)\big)\big\|_2$$
$$\leqslant \|\mathbb{E}'[y|x] - \bar{g}'(x)\|_2 + \|\mathbb{E}[y|x] - \bar{g}(x)\|_2$$
$$\leqslant \sigma_\mu^2 B_{f^{-1}}'^2 \big(2B_{\log p}' B_g' + B_g'' + 3dB_{f^{-1}}' B_f'' B_g'\big).$$

So for any $(s, v) \in \mathcal{S} \times \mathcal{V}$, by denoting $x := f(s, v)$, we have:

$$\big\|g(s) - g'(\Phi^{\mathcal{S}}(s, v))\big\|_2 = \big\|g((f^{-1})^{\mathcal{S}}(x)) - g'((f'^{-1})^{\mathcal{S}}(f(s, v)))\big\|_2 = \|\bar{g}(x) - \bar{g}'(x)\|_2$$
$$\leqslant \sigma_\mu^2 B_{f^{-1}}'^2 \big(2B_{\log p}' B_g' + B_g'' + 3dB_{f^{-1}}' B_f'' B_g'\big).$$

So the requirement is satisfied.

For requirement **(iii)**, note from the proof of Proposition B.2 that when $g$ is bijective and its inverse is $\frac{1}{2}$-Lipschitz, requirement **(ii)** implies requirement **(iii)**. So this $\Phi$ is a homeomorphism that makes $p$ $\delta$-semantic-dependent to $p'$ for any $\delta \geqslant \sigma_\mu^2 B_{f^{-1}}'^2 \big(2B_{\log p}' B_g' + B_g'' + 3dB_{f^{-1}}' B_f'' B_g'\big)$. $\square$

Note that although the $\delta$-semantic-dependency does not have transitivity, the above theorem is still informative: for any two CSGs sharing the same data distribution, particularly for a well-learned CSG $p$ and the ground-truth CSG $p^*$, the likelihood conversion error $\sup_{(s,v) \in \mathcal{S} \times \mathcal{V}} \big\|g(s) - g'(\Phi^{\mathcal{S}}(s, v))\big\|_2$, and the degree of mixing $v$ into $s$, measured by $\sup_{v^{(1)}, v^{(2)} \in \mathcal{V}} \big\|\Phi^{\mathcal{S}}(s, v^{(1)}) - \Phi^{\mathcal{S}}(s, v^{(2)})\big\|_2$, are bounded by $\sigma_\mu^2 B_{f^{-1}}'^2 \big(2B_{\log p}' B_g' + B_g'' + 3dB_{f^{-1}}' B_f'' B_g'\big)$.

## C More Explanations on the Model

**Explanations on our perspective.** We see the data generation process as generating the conceptual latent factors $(s, v)$ first, and then generating both $x$ and $y$ based on the factors. This follows Peters et al. (2017, Section 1.4) who promote the generation of an OCR dataset as the writer first comes up with an intension to write a character, and then writes down the character and gives its label based on the intension. It is also natural for medical image datasets, where the label may be diagnosed based on more fundamental features (*e.g.*, PCR test results showing the pathogen) that are not included in the dataset but actually cause the medical image. This generation process is also considered by Mcauliffe & Blei (2008); Kilbertus et al. (2018); Teshima et al. (2020).

On the labeling process from images that one would commonly think of, we also view it as a $s \to y$ process. Human directly knows the critical semantic feature $s$ (*e.g.*, the shape and position of each

stroke) by seeing the image, through the nature gift of the vision system (Biederman, 1987). The label is given by processing the feature (*e.g.*, the angle between two linear strokes, the position of a circular stroke relative to a linear stroke), which is a $s \to y$ process.

The causal graph in Fig. 1 implies that $x \perp\!\!\!\perp y|s$. This does not indicate that the semantic factor $s$ generates an image $x$ regardless of the label $y$. Given $s$, the generated image is dictated to hold the given semantics regardless of randomness, so the statistical independence does not mean semantic irrelevance. If an image $x$ is given, the corresponding label is given by $p(y|x)$, which is $\int p(s|x)p(y|s)\mathrm{d}s$ by the causal graph. So the semantic concept to cause the label through $p(y|s)$, is inferred from the image through $p(s|x)$.

**Comparison with the graph** $y_{\mathrm{tx}} \to s \to x \to y_{\mathrm{rx}}$**.** This graph is considered by one of our reviewers, under the perspective of a communication channel, where $y_{\mathrm{tx}}$ is a transmitted signal and $y_{\mathrm{rx}}$ is the received.

If the observed label $y$ is treated as $y_{\mathrm{tx}}$, the graph then implies $y \to s$. This is argued at the end of item **(2)** in Section 3 that it may make unreasonable implications. Moreover, the graph also implies that $y$ is a cause of $x$, as is challenged in item **(1)** in Section 3. The unnatural implications arise since intervening $y$ is different from intervening the "ground-truth" label. We consider $y$ as an observation that may be noisy, while the "ground-truth label" is never observed: one cannot tell if the labels at hand are noise-corrupted, based on the dataset alone. For example, the label of either image in Fig. 2 may be given by a labeler's random guess. Our adopted causal direction $s \to y$ is consistent with these examples and is also argued and adopted by Mcauliffe & Blei (2008); Peters et al. (2017, Section 1.4); Kilbertus et al. (2018); Teshima et al. (2020).

If the observed label $y$ is treated as $y_{\mathrm{rx}}$, the graph then implies $x \to y$, as is challenged in item **(1)** in Section 3. It is also argued by Schölkopf et al. (2012); Peters et al. (2017, Section 1.4); Kilbertus et al. (2018). Treating the observed label $y$ as $y_{\mathrm{rx}}$ and $y_{\mathrm{tx}}$ as the "ground-truth" label may be the motivation of this graph. But the graph implies that $y_{\mathrm{tx}} \perp\!\!\!\perp y_{\mathrm{rx}}|x$, that is, $p(y_{\mathrm{tx}}|x, y_{\mathrm{rx}}) = p(y_{\mathrm{tx}}|x)$ and $p(y_{\mathrm{rx}}|x, y_{\mathrm{tx}}) = p(y_{\mathrm{rx}}|x)$. So modeling $y_{\mathrm{tx}}$ (resp. $y_{\mathrm{rx}}$) does not benefit predicting $y_{\mathrm{rx}}$ (resp. $y_{\mathrm{tx}}$) from $x$.

# D   RELATION TO EXISTING DOMAIN ADAPTATION THEORY

**Existing DA theory**   In existing DA literature, the objective is to find a labeling function $h : \mathcal{X} \to \mathcal{Y}$ within a hypothesis space $\mathcal{H}$ that minimizes the target-domain risk $\tilde{R}(h) := \mathbb{E}_{\tilde{p}(x,y)}[\ell(h(x), y)]$ defined with a loss function $\ell : \mathcal{Y} \times \mathcal{Y} \to \mathbb{R}$. Since $\tilde{p}(x, y)$ is unavailable, it is of practical interest to consider the source-domain risk $R(h)$ and investigate its relation to $\tilde{R}(h)$. Ben-David et al. (2010a) give a bound relating the two risks:

$$\tilde{R}(h) \leqslant R(h) + 2d_1(p_x, \tilde{p}_x)$$
$$+ \min\{\mathbb{E}_{p(x)}[|h^*(x) - \tilde{h}^*(x)|], \mathbb{E}_{\tilde{p}(x)}[|h^*(x) - \tilde{h}^*(x)|]\}, \tag{43}$$
$$\text{where: } d_1(p_x, \tilde{p}_x) := \sup_{X \in \mathscr{X}} |p_x[X] - \tilde{p}_x[X]|.$$

Here $\mathscr{X}$ denotes the $\sigma$-algebra on $\mathcal{X}$, $d_1(p_x, \tilde{p}_x)$ is the *total variation* between the two distributions, and $h^* \in \operatorname{argmin}_{h \in \mathcal{H}} R(h)$ and $\tilde{h}^* \in \operatorname{argmin}_{\tilde{h} \in \mathcal{H}} \tilde{R}(\tilde{h})$ are the oracle/ground-truth labeling functions on the source and target domains, respectively (*e.g.*, $h^*(x) = \mathbb{E}[y|x]$ and $\tilde{h}^*(x) = \tilde{\mathbb{E}}[y|x]$ if $\operatorname{supp}(p_x) = \operatorname{supp}(\tilde{p}_x)$). Zhao et al. (2019) give a similar bound in the case of binary classification, in terms of the $\tilde{\mathcal{H}}$-divergence $d_{\tilde{\mathcal{H}}}$ in place of the total variance $d_1$, where $\tilde{\mathcal{H}} := \{\operatorname{sign}(|h(x) - h'(x)| - t) : h, h' \in \mathcal{H}, t \in [0, 1]\}$.

Ben-David et al. (2010a) also argue that in this bound, the total variation $d_1$ is overly strict and hard to estimate, so they develop another bound which is better known (asymptotically; omitting estimation error from finite samples):

$$\tilde{R}(h) \leqslant R(h) + d_{\mathcal{H}\Delta\mathcal{H}}(p_x, \tilde{p}_x) + \lambda_{\mathcal{H}}, \tag{44}$$
$$\text{where: } d_{\mathcal{H}\Delta\mathcal{H}}(p_x, \tilde{p}_x) := \sup_{h,h' \in \mathcal{H}} \left| \mathbb{E}_{p(x)}[\ell(h(x), h'(x))] - \mathbb{E}_{\tilde{p}(x)}[\ell(h(x), h'(x))] \right|,$$

$$\lambda_{\mathcal{H}} := \inf_{h \in \mathcal{H}} \left[ R(h) + \tilde{R}(h) \right].$$

Here $d_{\mathcal{H}\Delta\mathcal{H}}(p_x, \tilde{p}_x)$ is the $\mathcal{H}\Delta\mathcal{H}$-*divergence* measuring the difference between $p(x)$ and $\tilde{p}(x)$ under the discriminative efficacy of the labeling function family $\mathcal{H}$, and $\lambda_{\mathcal{H}}$ is the *ideal joint risk* achieved by $\mathcal{H}$. Long et al. (2015) give a similar bound in terms of maximum mean discrepancy (MMD) $d_K$ in place of $d_{\mathcal{H}\Delta\mathcal{H}}$.

For successful adaptation, DA often makes the *covariate shift assumption*: $\tilde{h}^* = h^*$ (or $p(y|x) = \tilde{p}(y|x)$) on $\operatorname{supp}(p_x, \tilde{p}_x) := \operatorname{supp}(p_x) \cup \operatorname{supp}(\tilde{p}_x)$.

**DA-DIR**  DA based on learning domain-invariant representations (DA-DIR) (Pan et al., 2010; Baktashmotlagh et al., 2013; Long et al., 2015; Ganin et al., 2016) aims to learn a deterministic representation extractor $\eta : \mathcal{X} \to \mathcal{S}$ to some representation space $\mathcal{S}$, in order to achieve a *domain-invariant representation* (DIR): $p(s) = \tilde{p}(s)$, where $p(s) := \eta_{\#}[p_x](s)$ and $\tilde{p}(s) := \eta_{\#}[\tilde{p}_x](s)$ are the representation distributions on the two domains. The motivation is that, if *DIR* is achieved, then the distribution difference term in bound Eq. (43) or Eq. (44) diminishes, thus the bound is hopefully tighter on the representation space $\mathcal{S}$ than on the original data space $\mathcal{X}$, so *minimizing the source risk* is more effective to minimizing the target risk. Let $g : \mathcal{S} \to \mathcal{Y}$ be a labeling function on the representation space. The end-to-end labeling function is effectively $h = g \circ \eta$. The typical objective for DA-DIR thus combines the two desiderata:

$$\min_{\eta \in \mathcal{E}, g \in \mathcal{G}} R(g \circ \eta) + d(\eta_{\#}[p_x], \eta_{\#}[\tilde{p}_x]),$$

where $d(\cdot, \cdot)$ is a metric or discrepancy ($d(q, p) = 0 \iff q = p$) on distributions, and $\mathcal{E}$ and $\mathcal{G}$ are the hypothesis spaces for $\eta$ and $g$, respectively.

For the *existence* of the solution of this problem, it is often assumed stronger that there exist $\eta^* \in \mathcal{E}$ and $g^* \in \mathcal{G}$ such that $\eta_{\#}^*[p_x] = \eta_{\#}^*[\tilde{p}_x]$ and $R(g^* \circ \eta^*) = R(h^*)$. *Assumption* 3 of Johansson et al. (2019) further assumes covariate shift and that $g^* \circ \eta^* = h^*$ on $\operatorname{supp}(p_x, \tilde{p}_x)$; that is, there exist $\eta^* \in \mathcal{E}$ and $g^* \in \mathcal{G}$ such that $\eta_{\#}^*[p_x] = \eta_{\#}^*[\tilde{p}_x]$ and $g^* \circ \eta^* = h^* = \tilde{h}^*$ on $\operatorname{supp}(p_x, \tilde{p}_x)$. They also mention that this is not guaranteed to hold in practice.

**Problems of DA-DIR**  Johansson et al. (2019); Zhao et al. (2019) give examples where even under as strong an assumption as Assumption 3 of Johansson et al. (2019) (*i.e.*, covariate shift and a strong existence assumption), the two desiderata of DA-DIR (*i.e.*, minimal source risk $R(g \circ \eta) = R(h^*)$ and DIR $\eta_{\#}[p_x] = \eta_{\#}[\tilde{p}_x]$) still allow the bounds to be uselessly loose and the target risk $\tilde{R}(g \circ \eta)$ to take its worst value (particularly, the two desiderata cannot guarantee $\eta = \eta^*$ or $g = g^*$ or $g \circ \eta = h^* = \tilde{h}^*$ on $\operatorname{supp}(p_x, \tilde{p}_x)$). This is essentially an identifiability problem.

The examples do not contradict existing DA bounds. Consider a given representation extractor $\eta$.

**(1)** Under Eq. (43). Applying the bound on the representation space $\mathcal{S}$ gives:

$$\begin{aligned}
\tilde{R}(g \circ \eta) &\leqslant R(g \circ \eta) + 2d_1(\eta_{\#}[p_x], \eta_{\#}[\tilde{p}_x]) \\
&\quad + \min\{\mathbb{E}_{\eta_{\#}[p_x](s)}[|g_{\eta}^*(s) - \tilde{g}_{\eta}^*(s)|], \mathbb{E}_{\eta_{\#}[\tilde{p}_x](s)}[|g_{\eta}^*(s) - \tilde{g}_{\eta}^*(s)|]\},
\end{aligned} \quad (45)$$

where $g_{\eta}^*$ and $\tilde{g}_{\eta}^*$ are the optimal labeling functions on top of the representation extractor $\eta$. In the covariate shift case, DIR $\eta_{\#}[p_x] = \eta_{\#}[\tilde{p}_x]$ and minimal source risk $R(g_{\eta}^* \circ \eta) = R(h^*)$ are not sufficient to guarantee $g_{\eta}^* = \tilde{g}_{\eta}^*$ (Ben-David et al., 2010b; Gong et al., 2016). Johansson et al. (2019) argue that they are still not sufficient even under their Assumption 3.

In both examples by Johansson et al. (2019); Zhao et al. (2019), the considered $\eta$, although achieving both desiderata, is not $\eta^*$, and this $\eta$ even renders different optimal $g$'s: $g_{\eta}^* \neq \tilde{g}_{\eta}^*$. Johansson et al. (2019) claim that it is necessary to require $\eta$ to be invertible to make $g_{\eta}^* = \tilde{g}_{\eta}^*$, and develop a bound that explicitly shows the effect of the invertibility of $\eta$. The $\eta$ in the examples is not invertible.

**(2)** Under Eq. (44). Applying the bound on the representation space $\mathcal{S}$ gives:

$$\begin{aligned}
\mathbb{E}_{\tilde{p}(s,y)}[\ell(g(s), y)] &\leqslant \mathbb{E}_{p(s,y)}[\ell(g(s), y)] + d_{\mathcal{G}\Delta\mathcal{G}}(\eta_{\#}[p_x], \eta_{\#}[\tilde{p}_x]) \\
&\quad + \inf_{g \in \mathcal{G}} \left[ \mathbb{E}_{\tilde{p}(s,y)}[\ell(g(s), y)] + \mathbb{E}_{p(s,y)}[\ell(g(s), y)] \right],
\end{aligned}$$

where $p_{s,y} := (\eta, \mathrm{id}_y)_{\#}[p_{x,y}]$ with $\mathrm{id}_y : (x,y) \mapsto y$ and similarly for $\tilde{p}_{s,y}$. Note that $\mathbb{E}_{p(s,y)}[\ell(g(s), y)] = \mathbb{E}_{p(x,y)}[\ell(g(\eta(x), y))] = R(g \circ \eta)$, so the last term on the r.h.s becomes: $\inf_{g \in \mathcal{G}} \left[ \tilde{R}(g \circ \eta) + R(g \circ \eta) \right] = \lambda_{\mathcal{G} \circ \eta}$, where $\mathcal{G} \circ \eta := \{g \circ \eta : g \in \mathcal{G}\}$. So the bound becomes:

$$\tilde{R}(g \circ \eta) \leqslant R(g \circ \eta) + d_{\mathcal{G} \Delta \mathcal{G}}(\eta_{\#}[p_x], \eta_{\#}[\tilde{p}_x]) + \lambda_{\mathcal{G} \circ \eta}. \tag{46}$$

This result is shown by Johansson et al. (2019). They argue that finding $\eta$ that achieves DIR and minimal training risk cannot guarantee a tighter bound since the last term $\lambda_{\mathcal{G} \circ \eta}$ may be very large.

In both examples by Johansson et al. (2019); Zhao et al. (2019), $\mathrm{supp}(p_x) \cap \mathrm{supp}(\tilde{p}_x) = \emptyset$. It may cause the problem that $g \circ \eta$ can be very different from $h^*$ on $\mathrm{supp}(\tilde{p}_x)$ even when $R(g \circ \eta) = R(h^*)$. The developed bound by Johansson et al. (2019) also explicitly shows the role of support overlap, thus can be called a support-invertibility bound. They also give an example to show that DIR (particularly implemented by minimizing MMD) is not necessary ("sometimes too strict") for learning the shared/invariant $p(y|x)$.

**(3) A third bound.** Zhao et al. (2019) develop another bound for binary classification, where $\mathcal{Y} := \{0, 1\}$ and $R(h) := \mathbb{E}_{p(x)}[|h^*(x) - h(x)|]$. Denote $d_{\mathrm{JS}}(p, q) := \sqrt{\mathrm{JS}(p, q)}$ as the JS distance (Endres & Schindelin, 2003), where $\mathrm{JS}(p, q)$ is the JS divergence. It is bounded: $0 \leqslant d_{\mathrm{JS}}(p, q) \leqslant 1$. It is shown that (Zhao et al., 2019, Lemma 4.8):

$$d_{\mathrm{JS}}(p_y, \tilde{p}_y) \leqslant d_{\mathrm{JS}}(\eta_{\#}[p_x], \eta_{\#}[\tilde{p}_x]) + \sqrt{R(g \circ \eta)} + \sqrt{\tilde{R}(g \circ \eta)}.$$

If $d_{\mathrm{JS}}(p_y, \tilde{p}_y) \geqslant d_{\mathrm{JS}}(\eta_{\#}[p_x], \eta_{\#}[\tilde{p}_x])$[9], it is shown that (Zhao et al., 2019, Theorem 4.3):

$$R(g \circ \eta) + \tilde{R}(g \circ \eta) \geqslant \frac{1}{2} \left( d_{\mathrm{JS}}(p_y, \tilde{p}_y) - d_{\mathrm{JS}}(\eta_{\#}[p_x], \eta_{\#}[\tilde{p}_x]) \right)^2, \tag{47}$$

or when the two domains are allowed to have their own representation-level labeling functions $g$ and $\tilde{g}$ (Zhao et al., 2019, Corollary 4.1),

$$R(g \circ \eta) + \tilde{R}(\tilde{g} \circ \eta) \geqslant \frac{1}{2} \left( d_{\mathrm{JS}}(p_y, \tilde{p}_y) - d_{\mathrm{JS}}(\eta_{\#}[p_x], \eta_{\#}[\tilde{p}_x]) \right)^2.$$

So when $p(y) \neq \tilde{p}(y)$, we have $d_{\mathrm{JS}}(p_y, \tilde{p}_y) > 0$, so DIR that minimizes $d_{\mathrm{JS}}(\eta_{\#}[p_x], \eta_{\#}[\tilde{p}_x])$ becomes harmful to minimizing the target risk $\tilde{R}(\tilde{g} \circ \eta)$.

Arjovsky et al. (2019) point out that in the covariate shift case, achieving a DIR $p(s) = \tilde{p}(s)$ implies $p(y) = \tilde{p}(y)$ (since $p(s) = \tilde{p}(s)$ and $p(y|s) = \tilde{p}(y|s)$). This may not hold in practice. When it does not hold, the bound above shows that DIR can limit prediction accuracy.

**Comparison with CSG**  Existing bounds Eqs. (43, 44, 45, 46) relate the source and target risks of a general and common labeling function $h \in \mathcal{H}$, *i.e.*, $\tilde{R}(h) - R(h)$, which is for bounding an objective; while our bound Eq. (36) relates the target risks of the optimal labeling functions on the source $h^*$ and target $\tilde{h}^*$ domains, *i.e.*, $\left| \tilde{R}(h^*) - \tilde{R}(\tilde{h}^*) \right|$ or $\left| \tilde{R}(h'^*) - \tilde{R}(\tilde{h}^*) \right|$, which measures the risk leap of the best source labeling function on the target domain. After adaptation, the prediction analysis (Eq. (42)) shows that CSG-DA achieves the optimal labeling function on the target domain in the infinite data limit.

For Eq. (47), we are not minimizing $d_{\mathrm{JS}}(\eta_{\#}[p(x)], \eta_{\#}[\tilde{p}(x)])$, so our method is good under that view. In fact, in our model the representation distributions on the two domains are $p(s) = \int p(s, v) \, \mathrm{d}v$ and $\tilde{p}(s) = \int \tilde{p}(s, v) \, \mathrm{d}v$ (replacing $\eta_{\#}[p(x)]$ and $\eta_{\#}[\tilde{p}(x)]$). We allow $p(s, v) \neq \tilde{p}(s, v)$ of course and do not seek to match them. Essentially, we do not rely on the invariance of $p(s|x)$ and $p(y|x)$, or $\eta^*$ and $h^*$, but the invariance of $p(x|s, v)$ in the other direction (generative direction). This thus allows $p(s|x) \neq \tilde{p}(s|x)$ and $p(y|x) \neq \tilde{p}(y|x)$, or $\eta^* \neq \tilde{\eta}^*$ and $h^* \neq \tilde{h}^*$, so we rely on an assumption different from the idea of all the bounds above. Since the data at hand is produced following a certain mechanism of nature anyway, the invariance in the generative direction $p(x|s, v)$ is thus more plausible (see Section 3.2).

---

[9]Unfortunately, it seems that the opposite direction holds when there exist $\eta^*$ and $g^*$ (unnecessarily the ones in the existence assumption or the Assumption 3 of Johansson et al. (2019)) such that: $p_y = (g^* \circ \eta^*)_{\#}[p_x]$ and $\tilde{p}_y = (g^* \circ \eta^*)_{\#}[\tilde{p}_x]$ and that $\eta$ is a reparameterization of $\eta^*$, due to the celebrated data processing inequality.

# E    METHODOLOGY DETAILS

## E.1    DERIVATION OF LEARNING OBJECTIVES

**The Evidence Lower BOund (ELBO).**    A common and effective approach to matching the data distribution $p^*(x, y)$ is to maximize likelihood, that is to maximize $\mathbb{E}_{p^*(x,y)}[\log p(x, y)]$. It is equivalent to minimizing $\mathrm{KL}(p^*(x, y)\|p(x, y))$ (note that $\mathbb{E}_{p^*}[\log p^*(x, y)]$ is a constant), so it drives $p(x, y)$ towards $p^*(x, y)$. But the likelihood function $p(x, y) = \int p(s, v, x, y) \, \mathrm{d}s\mathrm{d}v$ involves an intractable integration, which is hard to estimate and optimize. To address this, the popular method of *variational expectation-maximization* (variational EM) introduces a tractable (has closed-form density function and easy to sample) distribution $q(s, v|x, y)$ of the latent variables given observed variables, and a lower bound of the likelihood function can be derived:

$$\log p(x, y) = \log \mathbb{E}_{p(s,v)}[p(s, v, x, y)] = \log \mathbb{E}_{q(s,v|x,y)}\left[\frac{p(s, v, x, y)}{q(s, v|x, y)}\right]$$

$$\geqslant \mathbb{E}_{q(s,v|x,y)}\left[\log \frac{p(s, v, x, y)}{q(s, v|x, y)}\right] =: \mathcal{L}_{q,p}(x, y),$$

where the inequality follows Jensen's inequality and the concavity of the log function. The function $\mathcal{L}_{q,p}(x, y)$ is thus called *Evidence Lower BOund* (ELBO). The tractable distribution $q(s, v|x, y)$ is called *variational distribution*, and is commonly instantiated by a standalone model (from the generative model) called an *inference model*. Moreover, we have:

$$\mathcal{L}_{q,p}(x, y) + \mathrm{KL}(q(s, v|x, y)\|p(s, v|x, y))$$

$$= \mathbb{E}_{q(s,v|x,y)}\left[\log \frac{p(s, v, x, y)}{q(s, v|x, y)}\right] + \mathbb{E}_{q(s,v|x,y)}\left[\log \frac{q(s, v|x, y)}{p(s, v|x, y)}\right]$$

$$= \mathbb{E}_{q(s,v|x,y)}\left[\log \frac{p(s, v, x, y)}{p(s, v|x, y)}\right] = \mathbb{E}_{q(s,v|x,y)}[\log p(x, y)]$$

$$= \log p(x, y),$$

so maximizing $\mathcal{L}_{q,p}(x, y)$ w.r.t $q$ is equivalent to (note that the r.h.s $\log p(x, y)$ is constant of $q$) minimizing $\mathrm{KL}(q\|p(s, v|x, y))$ which drives $q$ towards the true posterior (*i.e.*, *variational inference*), and once this is (perfectly) done, $\mathcal{L}_{q,p}(x, y)$ becomes a lower bound of $\log p(x, y)$ that is tight at the current model $p$, so maximizing $\mathcal{L}_{q,p}(x, y)$ w.r.t $p$ effectively maximizes $\log p(x, y)$, *i.e.*, serves as maximizing likelihood. So the training objective becomes the expected ELBO, $\mathbb{E}_{p^*(x,y)}[\mathcal{L}_{q,p}(x, y)]$. Optimizing it w.r.t $q$ and $p$ alternately drives $p(x, y)$ towards $p^*(x, y)$ and $q(s, v|x, y)$ towards $p(s, v|x, y)$ eventually. The derivations and conclusions above hold for general latent variable models, with $(s, v)$ representing the latent variables, and $(x, y)$ observed variables (data variables).

**Variational EM for CSG.**    In the supervised case, the expected ELBO objective $\mathbb{E}_{p^*(x,y)}[\mathcal{L}_{q,p}(x, y)]$ can also be understood as the conventional supervised learning loss, *i.e.* the cross entropy, regularized by a generative reconstruction term. As explained in the main text (Section 4), after training, we only have the model $p(s, v, x, y)$ and an approximation $q(s, v|x, y)$ to the posterior $p(s, v|x, y)$, and prediction using $p(y|x)$ is still intractable. So we employ a tractable distribution $q(s, v, y|x)$ to model the required variational distribution as $q(s, v|x, y) = q(s, v, y|x)/q(y|x)$, where $q(y|x) = \int q(s, v, y|x) \, \mathrm{d}s\mathrm{d}v$ is the derived marginal distribution of $y$ (we will show that it can be effectively estimated and sampled from). With this instantiation, the expected ELBO becomes:

$$\mathbb{E}_{p^*(x,y)}[\mathcal{L}_{q,p}(x, y)]$$

$$= \int p^*(x, y)\frac{q(s, v, y|x)}{q(y|x)} \log \frac{p(s, v, x, y)q(y|x)}{q(s, v, y|x)} \, \mathrm{d}s\mathrm{d}v\mathrm{d}x\mathrm{d}y$$

$$= \int p^*(x, y)\frac{q(s, v, y|x)}{q(y|x)} \log q(y|x) \, \mathrm{d}s\mathrm{d}v\mathrm{d}x\mathrm{d}y + \int p^*(x, y)\frac{q(s, v, y|x)}{q(y|x)} \log \frac{p(s, v, x, y)}{q(s, v, y|x)} \, \mathrm{d}s\mathrm{d}v\mathrm{d}x\mathrm{d}y$$

$$= \int p^*(x)\left(\int p^*(y|x)\frac{\int q(s, v, y|x) \, \mathrm{d}s\mathrm{d}v}{q(y|x)} \log q(y|x) \, \mathrm{d}y\right)\mathrm{d}x$$

$$+ \int p^*(x)\left(\int \frac{p^*(y|x)}{q(y|x)}q(s, v, y|x) \log \frac{p(s, v, x, y)}{q(s, v, y|x)} \, \mathrm{d}s\mathrm{d}v\mathrm{d}y\right)\mathrm{d}x$$

$$= \mathbb{E}_{p^*(x)} \mathbb{E}_{p^*(y|x)} [\log q(y|x)] + \mathbb{E}_{p^*(x)} \mathbb{E}_{q(s,v,y|x)} \left[ \frac{p^*(y|x)}{q(y|x)} \log \frac{p(s,v,x,y)}{q(s,v,y|x)} \right],$$

which is Eq. (1). The first term is the (negative) expected cross entropy loss, which drives the inference model (predictor) $q(y|x)$ towards $p^*(y|x)$ for $p^*(x)$-a.e. $x$. Once this is (perfectly) done, the second term becomes $\mathbb{E}_{p^*(x)} \mathbb{E}_{q(s,v,y|x)} [\log p(s,v,x,y)/q(s,v,y|x)]$ which is the expected ELBO $\mathbb{E}_{p^*(x)} [\mathcal{L}_{q(s,v,y|x),p}(x,y)]$ for $q(s,v,y|x)$. It thus drives $q(s,v,y|x)$ towards $p(s,v,y|x)$ and $p(x)$ towards $p^*(x)$. It accounts for a regularization by fitting the input distribution $p^*(x)$ and align the inference model (predictor) with the generative model.

The target of $q(s,v,y|x)$, *i.e.* $p(s,v,y|x)$, adopts a factorization $p(s,v,y|x) = p(s,v|x)p(y|s)$ due to the graphical structure (Fig. 1) of CSG (*i.e.*, $y \perp\!\!\!\perp (x,v)|s$). The factor $p(y|s)$ is known (the invariant causal mechanism to generate $y$ in CSG), so we only need to employ an inference model $q(s,v|x)$ for the intractable factor $p(s,v|x)$, so $q(s,v,y|x) = q(s,v|x)p(y|s)$. Using this relation, we can reformulate Eq. (1) as:

$$\mathbb{E}_{p^*(x,y)} [\mathcal{L}_{q,p}(x,y)]$$

$$= \mathbb{E}_{p^*(x,y)} [\log q(y|x)] + \mathbb{E}_{p^*(x)} \left[ \int q(s,v|x)p(y|s) \frac{p^*(y|x)}{q(y|x)} \log \frac{p(s,v,x)}{q(s,v|x)} \, \mathrm{d}s\mathrm{d}v\mathrm{d}y \right]$$

$$= \mathbb{E}_{p^*(x,y)} [\log q(y|x)] + \mathbb{E}_{p^*(x)} \left[ \int \frac{p^*(y|x)}{q(y|x)} \left( \int q(s,v|x)p(y|s) \log \frac{p(s,v,x)}{q(s,v|x)} \, \mathrm{d}s\mathrm{d}v \right) \mathrm{d}y \right]$$

$$= \mathbb{E}_{p^*(x,y)} [\log q(y|x)] + \mathbb{E}_{p^*(x,y)} \left[ \frac{1}{q(y|x)} \mathbb{E}_{q(s,v|x)} \left[ p(y|s) \log \frac{p(s,v,x)}{q(s,v|x)} \right] \right],$$

which is Eq. (2). With this form of $q(s,v,y|x) = q(s,v|x)p(y|s)$, we have $q(y|x) = \mathbb{E}_{q(s,v|x)} [p(y|s)]$ which can also be estimated and optimized using reparameterization. For prediction, we can sample from the approximation $q(y|x)$ instead of the intractable $p(y|x)$. This can be done by ancestral sampling: first sample $(s,v)$ from $q(s,v|x)$, and then use the sampled $s$ to sample $y$ from $p(y|s)$.

The conclusions and methods can also be applied to general latent generative models for supervised learning, with $(s,v)$ representing the latent variables. When a model does not distinguish the two (groups of) latent factors $s$ and $v$ and treats them as one latent variable $z = (s,v)$, following a similar deduction gives:

$$\mathbb{E}_{p^*(x,y)} [\mathcal{L}_{q,p}(x,y)] = \mathbb{E}_{p^*(x,y)} [\log q(y|x)] + \mathbb{E}_{p^*(x,y)} \left[ \frac{1}{q(y|x)} \mathbb{E}_{q(z|x)} \left[ p(y|z) \log \frac{p(z,x)}{q(z|x)} \right] \right], \quad (48)$$

where $q(y|x) = \mathbb{E}_{q(z|x)} [p(y|z)]$. This is the conventional supervised variational auto-encoder (sVAE) baseline in our experiments.

**Variational EM to learn CSG with independent prior (CSG-ind).** See the main text in Section 4.1 for motivation and basic methods. Since the prior is the only difference between $p(s,v,x,y)$ and $p^{\perp\!\!\!\perp}(s,v,x,y)$, we have $p(s,v,x,y)/p^{\perp\!\!\!\perp}(s,v,x,y) = p(s,v)/p^{\perp\!\!\!\perp}(s,v) = p(s,v)/p(s)p(v) = p(v|s)/p(v)$. So $p(s,v,y|x) = \frac{p(v|s)}{p(v)} \frac{p^{\perp\!\!\!\perp}(x)}{p(x)} p^{\perp\!\!\!\perp}(s,v,y|x)$. As explained, inference models now only need to approximate the posterior $(s,v)|x$. Since $p(s,v,y|x) = p(s,v|x)p(y|s)$ and $p^{\perp\!\!\!\perp}(s,v,y|x) = p^{\perp\!\!\!\perp}(s,v|x)p(y|s)$ share the same $p(y|s)$ factor, we have $p(s,v|x) = \frac{p(v|s)}{p(v)} \frac{p^{\perp\!\!\!\perp}(x)}{p(x)} p^{\perp\!\!\!\perp}(s,v|x)$. The variational distributions $q(s,v|x)$ and $q^{\perp\!\!\!\perp}(s,v|x)$ target $p(s,v|x)$ and $p^{\perp\!\!\!\perp}(s,v|x)$ respectively, so we can express the former with the latter:

$$q(s,v|x) = \frac{p(v|s)}{p(v)} \frac{p^{\perp\!\!\!\perp}(x)}{p(x)} q^{\perp\!\!\!\perp}(s,v|x).$$

Once $q^{\perp\!\!\!\perp}(s,v|x)$ achieves its goal, such represented $q(s,v|x)$ also does so. So we only need to construct an inference model for $q^{\perp\!\!\!\perp}(s,v|x)$ and optimize it. With this representation, we have:

$$q(y|x) = \mathbb{E}_{q(s,v|x)} [p(y|s)] = \mathbb{E}_{q^{\perp\!\!\!\perp}(s,v|x)} \left[ \frac{p(v|s)}{p(v)} \frac{p^{\perp\!\!\!\perp}(x)}{p(x)} p(y|s) \right] = \frac{p^{\perp\!\!\!\perp}(x)}{p(x)} \mathbb{E}_{q^{\perp\!\!\!\perp}(s,v|x)} \left[ \frac{p(v|s)}{p(v)} p(y|s) \right]$$

$$= \frac{p^{\perp\!\!\!\perp}(x)}{p(x)} \pi(y|x),$$

where $\pi(y|x) := \mathbb{E}_{q^{\perp\!\!\!\perp}(s,v|x)}\left[\frac{p(v|s)}{p(v)}p(y|s)\right]$ as in the main text, which can be estimated and optimized using the reparameterization of $q^{\perp\!\!\!\perp}(s,v|x)$. From Eq. (2), the expected ELBO training objective can be reformulated as:

$$\mathbb{E}_{p^*(x,y)}[\mathcal{L}_{q,p}(x,y)]$$

$$= \mathbb{E}_{p^*(x,y)}\left[\log q(y|x) + \frac{1}{q(y|x)}\mathbb{E}_{q(s,v|x)}\left[p(y|s)\log\frac{p(s,v,x)}{q(s,v|x)}\right]\right]$$

$$= \mathbb{E}_{p^*(x,y)}\left[\log\frac{p^{\perp\!\!\!\perp}(x)}{p(x)} + \log\pi(y|x)\right.$$
$$\left. + \frac{p(x)}{p^{\perp\!\!\!\perp}(x)}\frac{1}{\pi(y|x)}\mathbb{E}_{q^{\perp\!\!\!\perp}(s,v|x)}\left[\frac{p(v|s)}{p(v)}\frac{p^{\perp\!\!\!\perp}(x)}{p(x)}p(y|s)\log\frac{p(s,v)p(x|s,v)}{\frac{p(v|s)}{p(v)}\frac{p^{\perp\!\!\!\perp}(x)}{p(x)}q^{\perp\!\!\!\perp}(s,v|x)}\right]\right]$$

$$= \mathbb{E}_{p^*(x,y)}\left[\log\frac{p^{\perp\!\!\!\perp}(x)}{p(x)} + \log\pi(y|x)\right.$$
$$\left. + \frac{1}{\pi(y|x)}\mathbb{E}_{q^{\perp\!\!\!\perp}(s,v|x)}\left[\frac{p(v|s)}{p(v)}p(y|s)\left(\log\frac{p(x)}{p^{\perp\!\!\!\perp}(x)} + \log\frac{p(s)p(v)p(x|s,v)}{q^{\perp\!\!\!\perp}(s,v|x)}\right)\right]\right]$$

$$= \mathbb{E}_{p^*(x,y)}\left[\log\frac{p^{\perp\!\!\!\perp}(x)}{p(x)} + \log\pi(y|x) + \frac{1}{\pi(y|x)}\mathbb{E}_{q^{\perp\!\!\!\perp}(s,v|x)}\left[\frac{p(v|s)}{p(v)}p(y|s)\right]\log\frac{p(x)}{p^{\perp\!\!\!\perp}(x)}\right.$$
$$\left. + \frac{1}{\pi(y|x)}\mathbb{E}_{q^{\perp\!\!\!\perp}(s,v|x)}\left[\frac{p(v|s)}{p(v)}p(y|s)\log\frac{p^{\perp\!\!\!\perp}(s,v)p(x|s,v)}{q^{\perp\!\!\!\perp}(s,v|x)}\right]\right]$$

$$= \mathbb{E}_{p^*(x,y)}\left[\log\frac{p^{\perp\!\!\!\perp}(x)}{p(x)} + \log\pi(y|x) + \frac{1}{\pi(y|x)}\pi(y|x)\log\frac{p(x)}{p^{\perp\!\!\!\perp}(x)}\right.$$
$$\left. + \frac{1}{\pi(y|x)}\mathbb{E}_{q^{\perp\!\!\!\perp}(s,v|x)}\left[\frac{p(v|s)}{p(v)}p(y|s)\log\frac{p^{\perp\!\!\!\perp}(s,v,x)}{q^{\perp\!\!\!\perp}(s,v|x)}\right]\right]$$

$$= \mathbb{E}_{p^*(x,y)}\left[\log\pi(y|x) + \frac{1}{\pi(y|x)}\mathbb{E}_{q^{\perp\!\!\!\perp}(s,v|x)}\left[\frac{p(v|s)}{p(v)}p(y|s)\log\frac{p^{\perp\!\!\!\perp}(s,v,x)}{q^{\perp\!\!\!\perp}(s,v|x)}\right]\right],$$

where in the second-last equality we have used the definition of $\pi(y|x)$. This gives Eq. (3). Note that $\pi(y|x)$ is not used in prediction, so there is no need to sample from it. Prediction is done by ancestral sampling from $q^{\perp\!\!\!\perp}(y|x)$, that is to first sample from $q^{\perp\!\!\!\perp}(s,v|x)$ and then from $p(y|s)$. Using this reformulation, we can train a CSG with independent prior even on data that manifests a correlated prior. The objective Eq. (5) on the training domain for domain adaptation can be derived similarly.

For numerical stability, we employ the log-sum-exp trick to estimate the expectations and compute the gradients.

### E.2 INSTANTIATING THE INFERENCE MODEL

Although motivated from learning a generative model, the method can be implemented using a general discriminative model (with hidden nodes) with causal behavior. By parsing some of the hidden nodes as $s$ and some others as $v$, a discriminative model could formalize a distribution $q(s,v,y|x)$, which implements the inference model and the generative mechanism $p(y|s)$. The parsing mode is shown in Fig. 3, which is based on the following consideration.

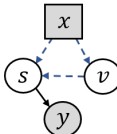

Figure 3: Parsing a general discriminative model as an inference model for CSG. The black solid arrow specifies $p(y|s)$ in the generative model, and the blue dashed arrows (representing computational directions but not causal directions) specify $q(s,v|x)$ (or $q^{\perp\!\!\!\perp}(s,v|x)$ or $\tilde{q}(s,v|x)$) as the inference model.

**(1)** The graphical structure of CSG in Fig. 1 indicates that $(v,x) \perp\!\!\!\perp y|s$, so the hidden nodes for $s$ should isolate $y$ from $v$ and $x$. The model then factorizes the distribution as $q(s,v,y|x) = q(s,v|x)q(y|s)$, and since the inference and generative models share the distribution on $y|s$ (see the main text for explanation), we can thus use the component $q(y|s)$ given by the discriminative model to implement the generative mechanism $p(y|s)$.

**(2)** The graphical structure in Fig. 1 also indicates that $s \not\perp\!\!\!\perp v|x$ due to the v-structure (collider) at $x$ ("explain away"). The component $q(s, v|x)$ should embody this dependence, so the hidden nodes chosen as $v$ should have an effect on those as $s$. Note that the arrows in Fig. 3 represent computation directions but not causal directions. We orient the computation direction $v \to s$ since all hidden nodes in a discriminative model eventually contribute to computing $y$.

After parsing, the discriminative model gives a mapping $(s, v) = \eta(x)$. We implement the distribution by[10] $q(s, v|x) = \mathcal{N}(s, v|\eta(x), \Sigma_q)$. For all the three cases of CSG, CSG-ind and CSG-DA, only one inference model for $(s, v)|x$ is required. The component $(s, v)|x$ of the discriminative model thus parameterizes $q^{\perp\!\!\!\perp}(s, v|x)$ and $\tilde{q}(s, v|x)$ for CSG-ind and CSG-DA. The expectations in all objectives (except for expectations over $p^*$ which are estimated by averaging over data) are all under the respective $(s, v)|x$. They can be estimated using $\eta(x)$ by the reparameterization trick (Kingma & Welling, 2014), and the gradients can be back-propagated.

We need two more components beyond the discriminative model to implement the method, *i.e.* the prior $p(s, v)$ and the generative mechanism $p(x|s, v)$. The latter can be implemented using a generator or decoder architecture comparable to the component $q(s, v|x)$. The prior can be commonly implemented using a multivariate Gaussian distribution, $p(s, v) = \mathcal{N}\left(\left(\begin{smallmatrix} s \\ v \end{smallmatrix}\right)\middle|\left(\begin{smallmatrix} \mu_s \\ \mu_v \end{smallmatrix}\right), \Sigma = \left(\begin{smallmatrix} \Sigma_{ss} & \Sigma_{sv} \\ \Sigma_{vs} & \Sigma_{ss} \end{smallmatrix}\right)\right)$. We parameterize $\Sigma$ via its Cholesky decomposition, $\Sigma = LL^\top$, where $L$ is a lower-triangular matrix with positive diagonals, which is in turn parameterized as $L = \left(\begin{smallmatrix} L_{ss} & 0 \\ M_{vs} & L_{vv} \end{smallmatrix}\right)$ with smaller lower-triangular matrices $L_{ss}$ and $L_{vv}$ and any matrix $M_{vs}$. Matrices $L_{ss}$ and $L_{vv}$ are parameterized by a summation of positive diagonals (guaranteed via an exponential map) and a lower-triangular (excluding diagonals) matrix. The conditional distribution $p(v|s)$ required by training CSG-ind is given by $p(v|s) = \mathcal{N}(v|\mu_{v|s}, \Sigma_{v|s})$, where $\mu_{v|s} = \mu_v + M_{vs}L_{ss}^{-1}(s - \mu_s)$, $\Sigma_{v|s} = L_{vv}L_{vv}^\top$ (see *e.g.*, Bishop (2006)). This prior does not imply a causal direction between $s$ and $v$ (the linear Gaussian case of Zhang & Hyvärinen (2009)) thus well serves as a prior for CSG.

## F    EXPERIMENT DETAILS

We use a validation set from the training domain for hyperparameter selection, to avoid overfitting to the finite training-domain data samples. The training and validation sets are constructed under a 80%-20% random split in each task. We note that hyperparameter selection in OOD tasks is itself controversial and nontrivial, and it is still an active research direction (You et al., 2019). It is argued that if a validation set from the test domain is available, a better choice would be to incorporate it in learning as the semi-supervised adaptation task, instead of using it just for validation. As our methods are designed to fit the training domain data and our theory shows guarantees under a good fit of the training-domain data distribution, hyperparameter selection using a training-domain validation set is reasonable.

We align the scale of the CE term in the objectives of all methods, and tune the coefficients of the ELBOs to be their largest values that make the final accuracy near 1 on the validation set, so that they wield the most power on the test domain while be faithful to explicit supervision. The coefficients are preferred to be large to well fit $p^*(x)$ (and $\tilde{p}^*(x)$ for domain adaptation) to gain generalizability in the test domain, while they should not affect training accuracy, which is required for a good fit of the training distribution.

The supervised variational auto-encoder (sVAE) baseline method is a counterpart of CSG that does not separate its latent variable $z$ into $s$ and $v$. This means that all its latent variables in $z$ directly (*i.e.*, not mediated by $s$) affect the output $y$. It is learned by optimizing Eq. (48) for OOD generalization, and adopts a similar method as CSG-DA for domain adaptation. To align the model architecture for fair comparison, this means that the latent variable $z$ of sVAE can only be taken as the latent variable $s$ in CSG.

All the experiments are implemented in PyTorch.

---

[10] Other approaches to introducing randomness are also possible, such as employing stochasticity on the parameters/weights as in Bayesian neural networks (Neal, 1995), or using dropout (Srivastava et al., 2014; Gal & Ghahramani, 2016). Here we adopt this simple treatment to highlight the main contribution.

### F.1 SHIFTED MNIST

We use a multilayer perceptron (MLP) with sigmoid activation with 784(for $x$)-400-200(first 100 for $v$)-50(for $s$ or $z$)-1(for $y$) nodes in each layer for the inference model of generative methods, and use an MLP with 50(for $s$)-(100(for $v$)+100)-400-784(for $x$) nodes in each layer for their generative component $p(x|s,v)$. We use a larger architecture with 784-600-300-75-1 nodes in each layer for discriminative methods to compensate additional parameters of generative methods. For all the methods, we use a mini-batch of size 128 in each optimization step, and use the RMSprop optimizer (Tieleman & Hinton, 2012), with weight decay parameter $1 \times 10^{-5}$, and learning rate $1 \times 10^{-3}$ for OOD generalization and $3 \times 10^{-4}$ for domain adaptation. These hyperparameters are chosen and then fixed, by running and then validating using CE and DANN. For generative methods, we take the Gaussian variances of $p(x|s,v)$ and $q(s,v|x)$ as $0.03^2$. The scale of the standard derivations of these conditional Gaussian distributions are chosen small to meet the intense causal mechanism assumption in our theory (*e.g.*, in Theorem 5.4).

We train the models for 100 epochs when all the methods converge in terms of loss and training accuracy. We align the scale of the CE term in the objectives of all methods, and scale the ELBO terms with the largest weight that makes training accuracy near 1 in OOD generalization. We then fix the tuned weight and scale the weight of adaptation terms in a similar way for domain adaptation. Other parameters are tuned similarly. For generative methods, the ELBO weight is $1 \times 10^{-5}$ selected from $\{1,3\} \times 10^{\{-6,-5\}} \cup 1 \times 10^{\{-2,-1,0,1,2\}}$, and the adaptation weights for sVAE-DA and CSG-DA are $1 \times 10^{-2}$ selected from $1 \times 10^{\{0,-1,-2,-3,-4\}}$ and $1 \times 10^{-5}$ selected from $1 \times 10^{\{0,-1,-2,-3,-4\}} \cup \{1,3\} \times 10^{\{-5,-6,-7,-8\}}$. For domain adaptation methods, the adaptation weight is $1 \times 10^{-4}$ except for CDAN which adopts $1 \times 10^{-2}$, all selected from $1 \times 10^{\{0,-1,-2,-3,-4\}}$. For CNBB, we use regularization coefficients $1 \times 10^{-4}$ and $3 \times 10^{-6}$ to regularize the sample weight and learned representation, and run 4 inner gradient descent iterations with learning rate $1 \times 10^{-3}$ to optimize the sample weight. These parameters are selected from a grid search where the range of the parameters are: $\{1,3\} \times 10^{\{-2,-3,-4\}}$, $\{1,3\} \times 10^{\{-4,-5,-6\}}$, $\{4,8\}$, $1 \times 10^{\{-1,-2,-3\}}$.

### F.2 IMAGECLEF-DA

We adopt the same setup as in Long et al. (2018). We use the ResNet50 structure pretrained on ImageNet as the backbone of the discriminative/inference model. Input images are cropped and resized to shape $(3, 224, 224)$. For CSG, we select the first 128 dimensions of the bottleneck layer (the resized last fully connected layer of ResNet50, with output dimension 1024) as the variable $v$, and the output of a subsequent fully connected layer with output dimension 1024 as the variable $s$. The output is produced by a linear layer built on $s$.

For generative methods (*i.e.*, our methods and sVAE(-DA)), we construct an image decoder/generator that uses the DCGAN model (Radford et al., 2015) pretrained on Cifar10 as the backbone. The pretrained DCGAN is adapted from the PyTorch-GAN-Zoo[11]. The generator connects to the DCGAN backbone by an MLP layer to match DCGAN's input dimension 120, and generates images of desired size $(3, 224, 224)$ by appending to DCGAN's output of size $(3, 64, 64)$ with an transposed convolution layer with kernel size 4, stride size 4 and padding size 16.

Following Long et al. (2018), we use a mini-batch of size 32 in each optimization step, and adopt the SGD optimizer with Nesterov momentum parameter 0.9, weight decay parameter $5 \times 10^{-4}$, and a shrinking step size scheme with initial scale $1 \times 10^{-3}$, shrinking exponent 0.75 and per-datum coefficient $6.25 \times 10^{-6}$. For CSG methods, the Gaussian variances of $p(x|s,v)$ and $q(s,v|x)$ are taken as 0.1 and 3.0, respectively. The ELBO weight is $1 \times 10^{-8}$ for CSG methods and $1 \times 10^{-7}$ for sVAE, both selected from $1 \times 10^{\{-2,-4,-6\}} \cup \{1,3\} \times 10^{\{-7,-8,-9,-10\}}$. The adaptation weights for sVAE-DA and CSG-DA are $1 \times 10^{-8}$ for task **C**→**P** and $1 \times 10^{-7}$ for task **P**→**C**, selected from the same range. For CNBB, we use regularization coefficients $1 \times 10^{-6}$ and $3 \times 10^{-6}$ to regularize the sample weight and learned representation, and run 4 inner gradient descent iterations with learning rate $1 \times 10^{-4}$ to optimize the sample weight. These parameters are selected from a grid search where the range of the parameters are: $1 \times 10^{\{-4,-5,-6,-7\}} \cup \{3 \times 10^{-6}\}$, $\{1,3\} \times 10^{\{-5,-6,-7\}}$, $\{4\}$, $1 \times 10^{\{-2,-3,-4,-5\}}$.

---

[11]https://github.com/facebookresearch/pytorch_GAN_zoo

