# OpenReview forum: "Learning Causal Semantic Representation for Out-of-Distribution Prediction"
_ICLR.cc/2021/Conference — Reject_

### Official Review · AnonReviewer2 · 2020-10-28
**Interesting causality based model and several theoretic results. Insufficient experiments and poor presentation**

**Rating:** 5
**Confidence:** 3

**Review:**

This paper proposes a Causal Semantic Generative model (CSG) to model semantic and variation factors separately and also provides a variational approach to learn the model.  It is proved that under some (perhaps strong) assumptions, CSG identifies the semantic factor on the training domain. Authors further show the boundedness of OOD generalization error based on the above result. Two  experiments are presented to validate the proposed method.

While the paper seems to have many interesting ideas and theoretic results, it is poorly presented and not well prepared, making it hard to evaluate the technical correctness. I also notice that the paper changes the original latex formatting by reducing much vertical space for almost all section titles, theorems, and equations. Also, the top margin of page 8 is heavily reduced. There is approximately 2/3 - 1 page more content than that of the original format with required page limit. Given this consideration, I also lower my score.

In summary, I recommend a rejection for the present version. I highly suggest the authors revise the paper to make it more readable, self-contained, and concise, and resubmit the paper to another top conference/journal.

Please find my questions/suggestions below.

1. The variables $s$, $v$, $x$, $y$ are introduced in the introduction part but then not mentioned when they are actually used in Section 3 to describe CSG. I suggest introducing the variables in Section 3.

2. Authors use many italic words, and more crucially, lots of mixed uses of italic and normal fonts, particularly in Section 3. It really affects the reading flow. For introducing the CSG part, I recommend firstly introduce the model and then use another few paragraphs to present the examples for illustration.

3. For the invariance principle 3.2, $p(s,v)$ is the only source of domain shift. Does it allow $p(v)$ and $p(s|v)$ (or $p(s)$ and $p(v|s)$) change while keeping $p(s,v)$ unchanged?

3. (a) Sometimes you use 'Theorem' and also 'Thm.' in the main text, please be consistent. (b) The places for citation also affects the reading flow. For example, right after Assumption 5.2,  'It is a common (Janzing et al., 2009; Shalit et al., 2017; Khemakhem et al., 2019; Lee et al., 2019) sufficient condition for the fundamental (Peters et al., 2014, Prop. 7) requirement of causal minimality
(Peters et al., 2014; 2017) for identifiability.' (c)  Some notations are not introduced, e.g.,  $q^{\indep}(x)$ in CSG-ind.

4. Why is it necessary to introduce CSG-ind? And in what case shall we consider CSG-ind? I did not find it well explained.

5.  Also for Assumption 5.2, 'It is a common sufficient condition for the fundamental requirement of causal minimality for identifiability.' However, causal minimality is not equivalent to that $f$ being bijective.

6. The paper should provide more details in comparing with IRM, and in fact I didn't get much information about the difference with IRM from Section 3.2.  Can the authors give more details about 'For noisy or degenerate mechanisms, ambiguity
occurs during inference (Fig. 2), and the inferred result notably relies on the prior.'? Also, IRM is not compared in the experiments.

7. In my opinion, the biggest contribution of this paper is to propose CSG, within which several theoretic results (identifiability, OOD error bound, etc.) are established under some necessary but maybe strong conditions. However, there is much less content on verifying that CSG does bring many benefits. One way is to conduct extensive experiments to verify so, but I find the experiments are not sufficient and also several methods (like IRM) are not included for comparison.

** after reading rebuttal**

Thanks for clarifications and an improved version. I decide to increase my evaluation to 5.

However, I think that the paper needs to take more content to illustrate the practical benefits of the proposed CSG frameworks, for the following reasons:
- the framework is proposed based on empirical observations like 'intervening an image by e.g. breaking a camera sensor unit when taking the image, does not change how the photographer labels it', which is not mathematically rigorous;
- the principles and assumptions are rather strong (though I understand one generally has to make assumptions in causality), and in practice it is not clear when such assumptions hold and how many applications satisfy these assumptions;
- the interesting derivations and theorems are also based on the CSG framework, which means, if the framework is incorrect, then these results may fail;
- the experiment settings are rather limited in the current version. I hope the authors to add further content in their next version, regardless of whether the paper gets accepted or rejected.

Lastly, I still feel it a bit tricky to change the original formatting in the previous submission.

---

> ### Author Response · Authors · 2020-11-19
> **Reply to AnonReviewer2**
>
> Thanks for your detailed comment and advice. We’ve recovered the standard spacing and modified the paper accordingly. Meanwhile, there may be some misunderstandings that we'd like to clarify.
> 1. The variables in Section 3 hold the same meaning as in the Introduction. The Introduction uses the symbols to make it easier to align the intuitions and motivations with the formal definitions in Section 3 .
> 2. We intended to use italic fonts to visually distinguish examples from the main context. We’ve unified the fonts in the updated paper based on your feedback.
> 3. We suppose that this question comes from a misunderstanding of our notation. Following the convention in machine learning, symbols $p(s)$, $p(v)$, $p(s|v)$ and $p(v|s)$ represent the marginal and conditional distributions of $p(s,v)$ (i.e., $p(s) = \int p(s,v) dv$, $p(v|s) = p(s,v)/p(s)$, etc.), but not standalone distributions, for which case we use a different symbol like $\tilde{p}(s)$.  So the change of $p(s,v)$ naturally allows the change of any of its marginals or conditionals as special cases, and if $p(s,v)$ is unchanged, then none of the factors shall change.
> 4. (a) & (b) We’ve modified the presentation accordingly. (c) We did not use the symbol $q^\perp(x)$. We introduced $q^\perp(s,v|x)$ in CSG-ind as the inference model in this case, and $p^\perp(x)$, $p^\perp(s,v|x)$, $p^\perp(s,v,x)$, $\mathbb{E}^\perp[y|x]$ are distributions (expectation) given by the CSG with the independent prior $p^\perp(s,v)$.
> 5. We explained in Paragraph “CSG-ind” in Section 4.1 that CSG-ind is an improved method for OOD generalization, which mitigates the spurious $s$-$v$ correlation in training. Formally, the independent prior has a larger entropy than the correlated prior, so it reduces the training-domain-specific information, and makes the model rely more on the causal mechanisms. Adjusting the prior is necessary to give a different result from inference invariance and better leverage causal invariance. CSG-ind also performs better than CSG in our experiments. We’ve made this clearer in the paragraph.
> 6. Indeed, causal minimality is not equivalent to that $f$ being bijective. Our statement claims that the bijectivity is sufficient for causal minimality and is commonly assumed in the literature. So, the statement does not contradict that fact. The following explanation shows the danger of violation but is not verifying the necessity (the violation allows but not guarantees the ambiguity). We’ve made the explanation clearer.
> 7. To understand the claim “for noisy or degenerate mechanisms, ambiguity occurs during inference (Fig. 2), and the inferred result notably relies on the prior”:
>   A reasonable generative mechanism for the image $x$ in Fig. 2(left) would give high values of $p(x|y)$ for $y=“3”$ or $y=“5”$ since other values of $y$ are not likely to generate such an $x$. So $p(x) \approx p(y=“3”) p(x|“3”) + p(y=“5”) p(x|“5”)$, and by Bayes rule, $p(y|x) \approx \frac{p(y) p(x|y)}{p(y=“3”) p(x|“3”) + p(y=“5”) p(x|“5”)}$ for $y=“3”$ or $“5”$ and $p(y|x) \approx 0$ for other values of $y$. If the prior shows $p(y=“3”) \gg p(y=“5”)$ (resp. "$\ll$"), then $p(y|x)$ would be peaked at $“3”$ (resp. $“5”$). So, the prior change across domains makes the inference rule change by nature, and there does not exist a domain-invariant inference model in this case. This poses a challenge on inference invariance.
>
>   Our methods CSG-ind (for OOD generalization) and CSG-DA (for domain adaptation) adjust the prior in the test domain. The new prior gives a different inference rule from that following inference invariance. The prior adjustment addresses the intrinsic change of the prediction rule across domains. We’ve made it clear in Sec. 3.2. Although the CSG method does not adapt the prior, causal invariance shows that its OOD generalization error is bounded (Theorem 5.5).
>
>   For the specific IRM method, it is not applicable to our settings since we only have __one__ training domain for both OOD generalization and domain adaptation tasks (last paragraph of Introduction). IRM tackles the domain generalization task, where there are multiple training domains. In our cases, the IRM algorithm degenerates to the conventional cross entropy (CE) optimization, which is already included as a baseline. We mention IRM for an example that follows inference invariance.
> 8. We believe that the theoretical results are also a way to verify the benefits of CSG, and the conditions are appropriate as we explained in the paper. We'd like to mention that our learning/adaptation methods are also a novel and significant contribution, which are delicately designed to ease both training and prediction and avoid multiple inference models, while faithfully following the first principle of variational Bayes.
>
>   For the experiments, IRM is not an applicable baseline, as we’ve explained above. We’re doing more experiments.

---

### Official Review · AnonReviewer4 · 2020-10-28
**By learning separate latent causal variables for both diversity and semantics, this work presents a tractable method to learn supervised models invariant to distributional shift.**

**Rating:** 7
**Confidence:** 3

**Review:**

The Causal Semantic Generative Model (CSG) presents an approach for learning both semantic and diverse latent causal variables in supervised settings using variational Bayes.  Contrary to many similar approaches which assume the label as the causal variable, this assumes a hidden causal variable which produces both the labels and observed features. Furthermore, this approach separates this latent causal variable into two components one for semantics, which impacts label generation and one for the diversity/variation which in combination with the semantic variable generates the observed features.

The authors present variations for learning this model which account for correlation/independence between semantic and diversity latent variables (CSG and CSG-ind) and also extend to settings where some data does contain labels (CSG-DA). The authors also show under which data generating assumptions the modeling holds and show how changes to the prior distribution of semantic and diversity variable distributions impact Out-of-Distribution Generalization error and Domain Adaptation Error.

The work presents empirical results demonstrating the effectiveness of this approach in both OOD settings (no test adaptation) and domain adaptation settings (test adaptation). For the presented experiments the authors show superior results in OOD generalization and competitive results in the domain adaptation setting. While the experiments present a compelling proof of concept, the tasks Shifted MNIST and ImageCLEF-DA are not the most representative challenges in their respective domains.  Would be interested to see performance in ColoredMNIST task for causal identification and OOD generalization as the generative structure is well understood as well as performance capabilities. The same could be said for the domain adaptation task, with Office-Home, VisDA-17, DomainNet or a variety of more challenging and representative tasks giving more empirical credibly to the experiments performed.

---

> ### Author Response · Authors · 2020-11-19
> **Reply to AnonReviewer4**
>
> Thanks for acknowledging the experimental results and the novelty of the model and theory. In addition, we explained how causal invariance is different from and more general than inference invariance that many existing works rely on, and showed how to build models and learning methods based on the principle. Moreover, our methods are also a novel and nontrivial contribution, which are convenient for both learning and prediction and avoid multiple inference models, while following the first principle of variational Bayes.
>
> For the experiments, we’d like to point out that Colored-MNIST is not suitable for our consideration. For both OOD generalization and domain adaptation, we require a _single_ training domain, while there are multiple training domains in Colored-MNIST. Merging the training domains loses information, making unfair comparisons. Moreover, in Colored-MNIST, different colors are represented as different positions in the tensor representation, so the challenge is similar to that in Shifted-MNIST. We’re doing more experiments.

---

### Official Review · AnonReviewer5 · 2020-11-05
**Strengths: Novel approach and theory. Weaknesses: Clarity and experiments**

**Rating:** 6
**Confidence:** 3

**Review:**

**Summary**
The paper focuses on the causal perspective of domain-generalization and domain adaptation setup for images. I.e. classifying an image under some distribution shift at test time. Similar to previous work [1-4], it assumes that some latent semantic-object representation (s) and semantic-domain representation (v) cause the image, and that these causal (generative) mechanisms (s,v ->x ) are stable, while their prior p(s,v) is prone to change at test time. It develops a new variational approach to estimate the generative distributions, and test the approach on two datasets for domain-generalization and domain-adaptation.

Overall, the paper suggests a novel approach and theory to an important problem. Its major weaknesses are in its clarity and the experimental part.

**Strong points**
Novelty: The paper provides a novel approach for estimating the likelihood of p(class|image), by developing a new variational approach for modelling the causal direction (s,v->x).
Correctness: Although I didn’t verify the details of the proofs, the approach seems technically correct. Note that I was not convinced that s->y (see weakness)

**Weak points**
Experiments and Reproducibility:
The experiments show some signal, but are not through enough:
• shifted-MNIST: it is not clear why shift=0 is much better than shift~$N(0,\sigma^2)$, since both cases incorporate a domain shift
• It would be useful to show the performance the model and baselines on test samples from the observational (in) distribution.
• Missing details about evaluation split for shifted-MNIST: Did the experiments used a validation set for hyper-param search with shifted-MNIST and ImageCLEF? Was it based on in-distribution data or OOD data?
• It would be useful to provide an ablation study, since the approach has a lot of "moving parts".
• It would be useful to have an experiment on an additional dataset, maybe more controlled than ImageCLEF, but less artificial than shifted-MNIST.
• What were the ranges used for hyper-param search? What was the search protocol?

Clarity:
• The parts describing the method are hard to follow, it will be useful to improve their clarity.
• It will be beneficial to explicitly state which are the learned parametrized distributions, and how inference is applied with them.
• What makes the VAE inference mappings (x->s,v) stable to domain shift? E.g. [1] showed that correlated latent properties in VAEs are not robust to such domain shifts.
• What makes v distinctive of s? Is it because y only depends on s?
• Does the approach uses any information on the labels of the domain?

Correctness: I was not convinced about the causal relation s->y. I.e. that the semantic concept cause the label, independently of the image. I do agree that there is a semantic concept (e.g. s) that cause the image. But then, as explained by [Arjovsky 2019] the labelling process is caused by the image. I.e. s->image->y, and not as argued by the paper. The way I see it, is like a communication channel: y_tx -> s -> image -> y_rx. Could the authors elaborate how the model will change if replacing s->y by y_tx->s ?


**Other comments:**
• I suggest discussing [2,3,4], which learned similar stable mechanisms in images.
• I am not sure about the statement that this work is the "first to identify the semantic factor and leverage causal invariance for OOD prediction" e.g. see [3,4]
• The title may be confusing. OOD usually refers to anomaly-detection, while this paper relates to domain-generalization and domain-adaptation.
• It will be useful to clarify that the approach doesn't use any external-semantic-knowledge.
• Section 3.2 - I suggest to add a first sentence to introduce what this section is about.
• About remark in page 6: (1) what is a deterministic s-v relation? (2) chairs can also appear in a workspace, and it may help to disentangle the desks from workspaces.

[1] Suter et al. 2018, Robustly Disentangled Causal Mechanisms: Validating Deep Representations for Interventional Robustness
[2] Besserve et al. 2020, Counterfactuals uncover the modular structure of deep generative models
[3] Heinze-Deml et al. 2017, Conditional Variance Penalties and Domain Shift Robustness
[4] Atzmon et al. 2020, A causal view of compositional zero-shot recognition





**EDIT: Post rebuttal**

I thank the authors for their reply. Although the authors answered most of my questions, I decided to keep the score as is, because I share similar concerns with R2 about the presentation, and because experiments are still lacking.

Additionally, I am concerned with one of the author's replies saying *All methods achieve accuracy 1 ... on the training distribution*, because usually there is a trade-off between accuracy on the observational distribution versus the shifted distribution (discussed by Rothenhäusler, 2018 [Anchor regression]): Achieving perfect accuracy on the observational distribution, usually means relying on the spurious correlations. And under domain-shift scenarios, this would hinder the performance on the shifted-distribution.

---

> ### Author Response · Authors · 2020-11-19
> **Reply to AnonReviewer5 (Part 1)**
>
> Thanks for acknowledging the importance, novelty and correctness, and providing the feedback and suggestions. We've updated the paper accordingly.
>
> Experiments:
> * By construction, training distributions of both classes still cover the position shift=0, and they cover equally dense thus do not introduce a bias. For shift\~$N(0,2^2)$, roughly half of the samples lie in the half field of the other class, where the training distribution of the other class covers more densely. So a classifier tends to be misled to the other class.
> * All methods achieve accuracy 1 up to \~.001 on a validation set from the training domain, which is the tuning criterion (above Sec. 6.1).
> * We used a validation set from the training domain, by a 20%-80% random split. If a validation set on the test domain is available, a better choice may be to incorporate it in learning. Our theory shows the guarantee based on a good fit of the training distribution, so validation on the training domain is reasonable.
> * All the model components (i.e., the prior, causal mechanisms and inference model) are required for a valid model. In the objective, ablating ELBO is the CE baseline, ablating the CE loss does not give comparable results (end of Sec. 4), and ablating the test-domain ELBO is CSG(-ind).
> * We’re doing more experiments.
> * For hyperparameters, “we align the scale of the CE term and tune their parameters to lie on the margin that makes the converging training accuracy near 1” (above Sec. 6.1). The coefficients of other terms in the loss are preferred to be large to well fit $p^*(x)$ (and $\tilde{p}^*(x)$) for generalizability, but they should not affect training accuracy for a good fit of training distribution. We’ve listed the ranges of search.
>
> Clarity:
> * We’ve polished the section. We’d love to hear details like what is ambiguous or needs more explanation.
> * We’ve made it explicit.
> * Regarding “stable inference mappings”, we’d mention that CSG-ind and CSG-DA give a different inference rule for the test domain since they use a different prior.
>
>   VAEs considered in [1] do not adapt the prior. Moreover, they do not see supervised data, and treat the latent factors symmetrically while we remove the $v\to y$ edge. Also, [1] considers disentanglement, which differs from identifying the semantics (2nd para. under Def. 5.3).
> * Yes it is (Remark (3) of Thm. 5.4; its proof). The better performance than sVAE also shows the effect of removing $v\to y$.
> * There is only _one_ training domain (last para., Sec. 1), so the methods do not use domain labels.
>
> On $s\to y$:
> * From our perspective:
>
>   The data generation process follows Peters et al. (2017, Sec. 1.4): to generate an OCR datum, a writer comes up with an intension to write a character, and then writes it down and gives its label. It is also natural for medical image datasets, where the label may be diagnosed based on more fundamental features (e.g., PCR test for pathogen) that cause the image but are hidden. We also verified the causal implications in Sec. 3.
>
>   We view the labeling process from images also as a $s\to y$ process. Human directly knows the critical semantic feature (e.g., the shape and position of each stroke) by seeing the image (Biederman, 1987). The label is given by processing the feature (e.g., the angle between two linear strokes, the position of a circular stroke relative to a linear stroke), which is a $s\to y$ process.
>
>   Our causal graph does not indicate “the semantic concept causes the label, independently of the image”. Given an image $x$, the label is given by $p(y|x)=\int p(s|x)p(y|s) ds$ by the causal graph. So the semantics to cause the label is inferred from the image through $p(s|x)$. If you meant $x\perp y|s$, the generated image is dictated to hold the given semantics regardless of randomness, so the independence does not mean semantic irrelevance.
> * Explanations under $y_{tx}\to s\to x\to y_{rx}$.
>   - Treating the observed label $y$ as $y_{tx}$:
>
>     The graph implies $y\to s$, but we argued that this unreasonably implies adding noise to the labels in a dataset automatically changes object features and consequently the images, and changing the object shape, texture, etc. to those of a different object does not change the label (end of item (2), Sec. 3). Note that we consider $y$ as an observation that may be noisy, while the “ground-truth” label is never observed (one cannot tell if the labels at hand are noisy; e.g., for either image in Fig. 2, the label may be a random guess by a labeler). The adopted $s\to y$ is consistent with the examples and is also supported by Mcauliffe & Blei (2008); Peters et al. (2017); Teshima et al. (2020).
>   - Treating $y$ as $y_{rx}$:
>
>     We suppose $y_{tx}$ is treated as the “ground-truth” label in this case. Since the graph implies $y_{tx}\perp y_{rx}|x$, modeling $y_{tx}$ (resp. $y_{rx}$) does not benefit predicting $y_{rx}$ (resp. $y_{tx}$) from $x$. It also implies $x\to y$, which we challenged in item (1), Sec. 3.

---

> ### Author Response · Authors · 2020-11-19
> **Reply to AnonReviewer5 (Part 2)**
>
> Other comments:
> * [2] is similar to [1]. It proposes definitions and metrics for studying the disentanglement of learned latent factors by existing deep generative models, and does not aim at learning stable mechanisms. [3, 4] are discussed below.
> * Although sharing similar ideas, [3, 4] do not faithfully follow the causal invariance principle and also rely on standalone intuitions, and resort to inference invariance. They do not show the identifiability guarantee of the semantics.
>
>   [3] requires observing the identity $i$ in an image $x$, and assumes the invariance of $p(s|y,i)$. But it does not indicate the invariance of $f(x) := \mathbb{E}[y|x]$ which is used in all domains, nor the small variance of $p(f(x)|y,i)$ which is to be minimized. We view the former to follow inference invariance, and the latter may have a large variance under causal invariance (Sec. 3.2; Fig. 2). It is a discriminative model and do not learn the causal mechanisms. The semantic variable is not present in the method.
>
>   [4]  “selects $(\hat{a}, \hat{o}) = \arg\max_{a,o} p(x|a,o)$ for inference”, which amounts to using the same prior (the uniform prior) in all domains thus follows inference invariance. The triplet loss and invertible embedding loss are also not from the causal graph. Our methods respect the graphical structure, so we do not need a standalone independence loss.
>
>   We've included these discussions in the updated paper.
> * As we understand it, “out-of-distribution” is not confined to “OOD detection”, i.e. anomaly detection. We consider “OOD prediction”, where test queries come from a different distribution than the training domain. It is also a common usage of OOD, e.g., Arjovsky et al. (2019) use it for the domain generalization task. Note that domain generalization requires multiple training domains, while the OOD generalization task we consider only requires one (last paragraph of Sec. 1).
> * Remark in page 6: (1) Formally, it refers to the case where $s$ is a function of $v$ and vice versa. (2) If chairs are also considered, then $s\in${desk, bed, chair} cannot be a function of $v\in${workspace, bedroom}. So, it does not count for a deterministic $s$-$v$ case, and satisfies the theorem condition for identifying $s$.

---

### Author Response · Authors · 2020-11-25
**Experiments updated**

Dear reviewers,

We conducted more experiments with a finer implementation. For generative methods (including sVAE(-DA) baseline), we employ a DCGAN model pretrained on Cifar10 (obtained from [PyTorch-GAN-Zoo](https://github.com/facebookresearch/pytorch_GAN_zoo)) as the backbone of the generator network. Improved performance is achieved on ImageCLEF C->P task. We also conducted experiments on the P->C task in the other direction, and our methods achieve the best OOD generalization performance and comparable performance with modern domain adaptation methods.

In all, implementation details considerably affect the empirical performance, and we are still seeking for more suitable implementation to wield the power the proposed methods.

Thanks again for your effort and feedback! Hope you can enjoy the new version of the paper.

---

### Decision · Program_Chairs · 2021-01-07
**Final Decision**

**Decision:**

Reject

**Comment:**

The paper formalizes domain adaptation by taking the causal (generative) direction of dependencies p(image | class, domain).  They evaluate an ELBO surrogate loss by fitting a reverse q function that is new for this setup, and add a term to the loss that induces independence between class and domain. The paper also  proves identifiability conditions. The approach is then evaluated on two semi-synthetic and small datasets, showing some improvement.

Reviewers were concerned about presentation and the experimental validation. The authors addresses some of the concerns in their rebuttal, but several reviewers found that the experimental evidence was still lacking, and that the authors should evaluate their approach in more standard and realistic benchmark datasets.  As a result, the paper cannot be accepted in its current form